# TOWARDS ONE-STEP CAUSAL VIDEO GENERATION VIA ADVERSARIAL SELF-DISTILLATION

**Yongqi Yang**[1,*]**Huayang Huang**[1,*]**, Xu Peng**[2,*]**, Xiaobin Hu**[2]**, Donghao Luo**[2,†]**,
Jiangning Zhang**[2]**, Chengjie Wang**[2]**, Yu Wu**[3†]
[1]School of Computer Science, Wuhan University    [2]Tencent YouTu Lab
[3]School of Artificial Intelligence, Wuhan University
{yongqiyang, hyhuang, wuyucs}@whu.edu.cn
Project page: https://github.com/BigAandSmallq/SAD.git

## ABSTRACT

Recent hybrid video generation models combine autoregressive temporal dynamics with diffusion-based spatial denoising, but their sequential, iterative nature leads to error accumulation and long inference times. In this work, we propose a distillation-based framework for efficient causal video generation that enables high-quality synthesis with extreme limited denoising steps. Our approach builds upon Distribution Matching Distillation (DMD) framework and proposes a novel form of Adversarial Self-Distillation (ASD) strategy, which aligns the outputs of the student model's $n$-step denoising process with its $(n+1)$-step version in the distribution level. This design provides smoother supervision by bridging small intra-student gaps and more informative guidance by combining teacher knowledge with locally consistent student behavior, substantially improving training stability and generation quality in extremely few-step scenarios. In addition, we present a First-Frame Enhancement (FFE) strategy, which allocates more denoising steps to the initial frames to mitigate error propagation while applying larger skipping steps to later frames. Extensive experiments on VBench demonstrate that our method surpasses state-of-the-art approaches in both one-step and two-step video generation. Notably, our framework produces a single distilled model that flexibly supports multiple inference-step settings, eliminating the need for repeated re-distillation and enabling efficient, high-quality video synthesis.

## 1 INTRODUCTION

Diffusion models (Ho et al., 2020; Song et al., 2021a; Nichol & Dhariwal, 2021) have achieved remarkable progress in high-quality image (Dhariwal & Nichol, 2021; Nichol et al., 2022; Rombach et al., 2022) and video generation (Ho et al., 2022; Brooks et al., 2024; Blattmann et al., 2023). However, their application to long-form video (Chen et al., 2023; Zhang et al., 2023) and interactive settings (Che et al., 2025) remains limited by the high computational cost of iterative denoising and the reliance on bidirectional attention, which requires synthesizing entire sequences jointly. Autoregressive methods (Liang et al., 2022; Ge et al., 2022; Wang et al., 2024), in contrast, allow causal frame-by-frame generation, but often suffer from error accumulation that compromises realism. Recent works (Chen et al., 2024; Jin et al., 2025; Li et al., 2025) have attempted to combine autoregressive temporal modeling with diffusion-based spatial refinement, yet these hybrid approaches inherit the efficiency bottleneck of multi-step denoising.

To alleviate this limitation, distillation has emerged as a promising direction for accelerating diffusion models (Sauer et al., 2024b; Yin et al., 2024b; Lu et al., 2025). By reducing a multi-step model to a few-step counterpart, distillation greatly improves efficiency while preserving generation quality. Nonetheless, existing distillation objective primarily focuses on aligning the prediction distributions of the few-step student with those of the multi-step teacher. When the student performs only 1- or 2-step generation, the discrepancy from the multi-step teacher becomes excessively large, making

---

*Equal contributions. Work done during an internship at Tencent YouTu Lab.
†Corresponding authors.

direct alignment unstable and causing severe quality degradation (Cheng et al., 2025; Yin et al., 2025; Huang et al., 2025). In other words, the fewer denoising steps, the larger the semantic and statistical gap to be bridged, which explains why extremely few-step distillation is particularly challenging.

In this paper, we introduce a novel method to address the challenge of high-quality video generation with minimal denoising steps (e.g. 1 and 2 steps). We extend the distribution matching distillation (DMD) framework (Yin et al., 2024b;a), and propose a novel Adversarial Self-Distillation (ASD) strategy. Unlike prior distillation methods (Song et al., 2023; Song & Dhariwal, 2024; Lin et al., 2024) that rely solely on supervision from the multi-step teacher, ASD provides the student with additional guidance from its own intermediate variants with slightly different denoising steps. Specifically, a discriminator is employed to adversarially match the $n$-step and $(n+1)$-step denoising distribution. As shown in Fig. 1, this step-wise self-alignment has two advantages: (1) it produces smoother supervision by bridging smaller step-to-step gaps

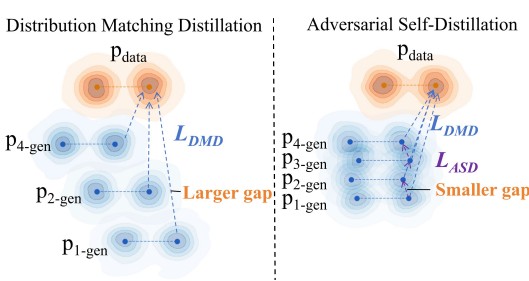

Figure 1: Conceptual illustration of the different alignment strategies of DMD and ASD during distillation. The points plotted in the figure indicate the multiple modes (or peaks) of the real data distribution, highlighting its multimodal nature.

instead of the large teacher–student gap, and (2) it yields more informative signals, since the student learns from both global teacher knowledge and its own locally consistent behavior. Together, these factors substantially enhance training stability and generation quality under extreme few-step settings. Fig. 2 also demonstrates that our method substantially reduces the performance degradation associated with a decreasing number of inference steps. Moreover, our framework introduces a **step-unified** design: instead of training a separate distilled model for each desired step size, a single student model trained with ASD can flexibly support multiple inference-step configurations at deployment. This property significantly improves practical usability, as it removes the need for repetitive re-distillation, which is especially valuable in scenarios such as dynamically balancing speed and quality trade-offs and accommodating deployment across varied resource settings.

To further improve video fidelity, we first conduct empirical analysis and observe that later frames exhibit higher generative redundancy compared to the first frame. Motivated by this, we propose a frame-wise inference strategy with varying denoising strengths. Unlike previous methods that treat all frames equally, we allocate more denoising steps to the crucial initial frames to mitigate error accumulation, while later frames are generated with larger skipping steps. This First-Frame Enhancement (FFE) strategy maintains a low overall computational cost while notably improving visual quality. Extensive experiments on VBench demonstrate that our method surpasses state-of-the-art approaches in both one-step and two-step video generation. Importantly, it achieves both efficiency and flexibility by using a single distilled model to support a wide range of inference settings.

In summary, our contributions are as follows:

- We propose an adversarial self-distillation strategy that aligns prediction distributions across different denoising steps of the student, significantly improving few-step generation quality.

- We propose a frame-wise inference strategy that allocates more denoising steps to crucial initial frames, reducing error accumulation and improving video fidelity.

- Our experiments demonstrate that our method surpasses the state-of-the-art in few-step generation quality while eliminating the need for separate distillation training for each desired step size.

## 2 RELATED WORK

### 2.1 AUTOREGRESSIVE/DIFFUSION VIDEO GENERATION

Diffusion (Blattmann et al., 2023; Ho et al., 2022; Yang et al., 2025) and autoregressive models (Ge et al., 2022; Kondratyuk et al., 2024; Yu et al., 2024) are the two dominant paradigms in video

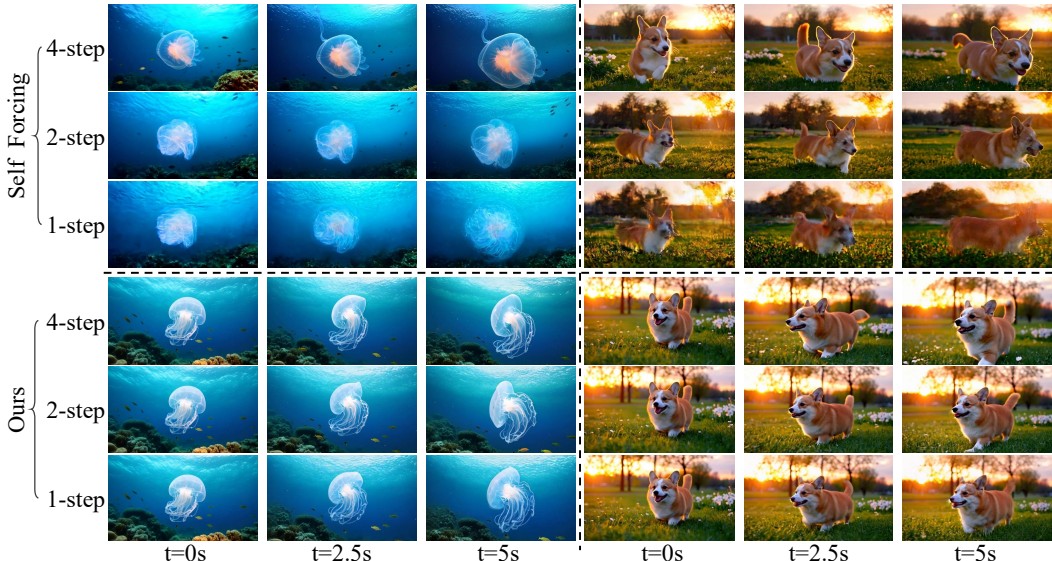

Figure 2: Qualitative results of Self Forcing (Huang et al., 2025) and ours under 4-step, 2-step and 1-step generation. Our method consistently maintains high-quality generation across 4-step, 2-step, and 1-step inference.

generation due to their ability to produce high-quality results. Diffusion models employ bidirectional attention (Bao et al., 2022; Peebles & Xie, 2023) to denoise and synthesize all frames simultaneously, achieving strong temporal consistency but preventing frame-wise editing or interactive generation. In contrast, autoregressive models generate frames sequentially by predicting the next token conditioned on previously generated frames. However, this strong dependence on earlier outputs often leads to error accumulation, making it difficult to synthesize highly realistic long videos (Che et al., 2025).

Recent works (Chen et al., 2024; Hu et al., 2025; Jin et al., 2025; Li et al., 2025; Yin et al., 2025; Weng et al., 2024) have proposed hybrid frameworks that integrate diffusion and autoregressive generation. A common design is to model temporal dynamics autoregressively while applying diffusion-based iterative denoising spatially. Despite their improved flexibility, their sequential and iterative nature introduces more complex generation processes and prolonged inference times. Building upon this line of research, our work adopts a distillation-based framework that significantly reduces generation latency, enhancing real-time interactivity in video synthesis.

## 2.2 ADVERSARIAL DISTILLATION

A line of work (Song et al., 2023; Song & Dhariwal, 2024; Liu et al., 2022; Meng et al., 2023; Berthelot et al., 2023) focuses on distilling the multi-step generation process into a few steps to improve the sampling efficiency of diffusion models. These methods train a student model to approximate the ordinary differential equation (ODE) trajectory of the original teacher model. Adversarial distillation uses a discriminator to align the distribution of the student model with the target distribution. ADD (Sauer et al., 2024b) and UFOGen (Xu et al., 2024) aim for the student model's final output to be indistinguishable from real data.

Further advancements have sought to preserve the generation trajectory itself, not just the final output. SDXL-Lightning (Lin et al., 2024) and LADD (Sauer et al., 2024a) achieve this by aligning the intermediate denoising states of the student model with those of the teacher model. Zhang et al. (2024) and Lin et al. (2025) fine-tune a pre-trained diffusion video model for AR generation via adversarial training, but they directly align one-step outputs with real data. Our work employs a discriminator to align the generated results of an $n$-step denoising process with those of an $(n+1)$-step process. This approach enhances the model's performance and consistency across various limited-step scenarios.

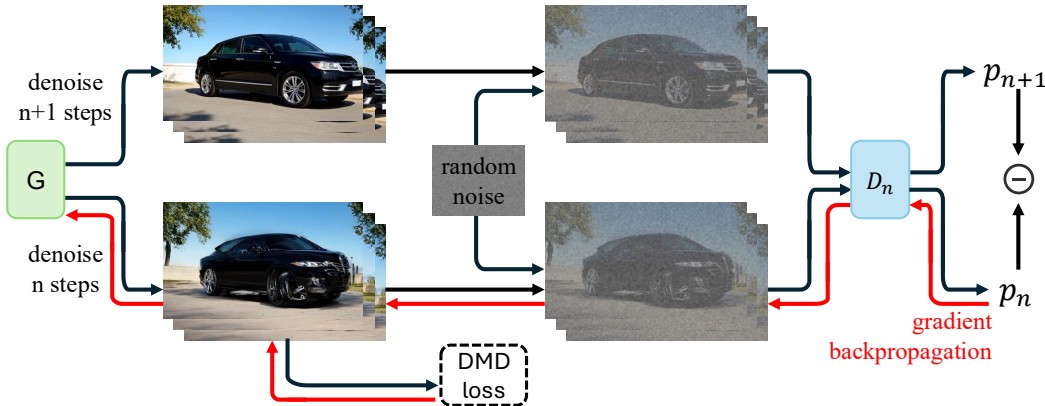

Figure 3: Pipeline of our proposed adversarial self-distillation process. We employ a discriminator $D_n$ to align the randomly noised $n$-step video with the $(n+1)$-step one through calculating the ASD loss in Equation (4). The generator $G$ is optimized using a combined objective function that includes the standard DMD loss and the ASD loss. Note that distillation is performed in the latent space, while the pixel domain is primarily used for visual analysis and display.

## 2.3 SCORE DISTILLATION

Existing methods (Zhou et al., 2024; Lu et al., 2025) utilize score-based models to achieve distribution matching across different noise states during distillation. For example, Dreamfusion (Poole et al., 2023) and ProlificDreamer (Wang et al., 2023) leverage a T2I diffusion model's score function to guide text-conditioned 3D generation. Distribution Matching Distillation (DMD) (Yin et al., 2024b;a) extends this approach to accelerate diffusion model generation by employing the original model as a score estimator for real data and a fine-tuned version as a score estimator for generated data.

Our approach is similar in that it aims to maintain a consistent sample probability across distributions under a particular noise level. To achieve this, we align the noisy version of the predictions from our few-step model's $n$-step and $(n+1)$-step denoising.

## 3 METHODOLOGY

To achieve high-quality and flexible causal video generation with minimal computational overhead, our distillation process employs two key strategies. We use Distribution Matching Distillation (DMD) to distill a multi-step generative model into a few-step student model through score-based alignment. Meantime, we introduce a novel form of self-supervision by aligning the student model's $n$-step denoising with its $(n+1)$-step version in the distribution level. This dual supervision not only enhances the generation quality but also enables the final model to produce consistent, hight-quality output across a variety of few-step settings. The few-step generator, the generator's score estimator and the discriminator are trained alternately to facilitate this process. The pipeline of the Adversarial Self-Distillation (ASD) process is depicted in Fig. 3. During inference, we further improve efficiency and quality by employing a First-Frame Enhancement (FFE) strategy, which uses a more intensive denoising process for the initial frame while reducing the number of steps for subsequent frames.

### 3.1 PRELIMINARY: DISTRIBUTION MATCHING DISTILLATION

Distribution Matching Distillation (DMD) (Yin et al., 2024a;b) presents a score-based approach to align a few-step student model with a multi-step teacher model. Unlike other distillation methods (Song et al., 2023; Song & Dhariwal, 2024), DMD not only accelerates generation but also improves output quality by aligning the output distributions. In diffusion models, the score function is represented by the gradient of the log probability of the distribution:

$$s_\theta(\boldsymbol{x}_t, t) = \nabla_{\boldsymbol{x}_t} \log p(\boldsymbol{x}_t) = -\frac{\epsilon_\theta(\boldsymbol{x}_t, t)}{\sigma_t}, \tag{1}$$

---

**Algorithm 1** Adversarial Self-Distillation Process

---

**Require:** Few-step denoising steps $\mathcal{T} = \{t_1, \ldots, t_N\}$, score function of teacher model $s_{\text{data}}$
 1: **Initialize** few-step student model $G_\theta$ with the original model
 2: **Initialize** generator's score function $s_{\text{gen}}$ with $s_{\text{data}}$ estimated by the original model
 3: **Initialize** discriminator $D_\psi^n$ with the original model with trainable heads, $D_\psi^n$ and $s_{\text{gen}}$ share the same backbone parameters
 4: **while** training **do**
 5: $\quad n \leftarrow \text{RandomInteger}(1, N)$
 6: $\quad$ **Predict** few-step student model generate sample $\boldsymbol{x}_0^{n+1} = G_\theta^{n+1}(\boldsymbol{z_1}), \boldsymbol{z_1} \sim \mathcal{N}(0, I)$
 7: $\quad$ **Add noise** $\boldsymbol{x}_t^1 = (1 - t)\boldsymbol{x}_0^{n+1} + t\epsilon, \ \epsilon \sim \mathcal{N}(0, I), t \sim \text{Uniform}(\mathcal{T})$
 8: $\quad$ **Predict** few-step student model generate sample $\boldsymbol{x}_0^n = G_\theta^n(\boldsymbol{z_2}), \boldsymbol{z_2} \sim \mathcal{N}(0, I)$
 9: $\quad$ **Add noise** $\boldsymbol{x}_t^2 = (1 - t)\boldsymbol{x}_0^n + t\epsilon, \ \epsilon \sim \mathcal{N}(0, I)$
10: $\quad$ **Update** few-step generator $G_\theta$ using DMD loss and ASD loss $\qquad\qquad\qquad$ ▷ Eq. 5
11: $\quad$ **Update** discriminator $D_\psi^n$ with ASD loss $\qquad\qquad\qquad\qquad\qquad\qquad$ ▷ Eq. 4
12: $\quad$ **Update** generator's score function $s_{\text{data}}$ using diffusion loss $\mathcal{L}_{\text{gen}}^\phi$ $\qquad\qquad$ ▷ Eq. 3
13: **end while**

---

where $\boldsymbol{x}_t$ is the noisy sample at timestep $t$ with $p(\boldsymbol{x}_t)$ as its corresponding data distribution, $t \in \{0, ..., T\}$. $p(\boldsymbol{x}_T)$ is a standard Gaussian distribution. $\epsilon_\theta(\boldsymbol{x}_t, t)$ is the predicted output of the generative model parameterized by $\theta$ and $\sigma_t$ is predefined by the noise schedule (Karras et al., 2022; Song et al., 2021b). DMD minimizes the reverse Kullback-Leibler (KL) divergence between the score of the true data distribution and the score of the student model's generated data distribution. The gradient of DMD loss can be represented by:

$$\nabla_\theta \mathcal{L}_{\text{DMD}} \triangleq \mathbb{E}_t \left( \nabla_\theta \text{KL} \left( p_{\text{gen},t} \| p_{\text{data},t} \right) \right)$$

$$\approx - \underset{\boldsymbol{z},t,\boldsymbol{x}_t}{\mathbb{E}} \left[ \left( s_{\text{data}} \left( \boldsymbol{x}_t, t \right) - s_{\text{gen}} \left( \boldsymbol{x}_t, t \right) \right) \frac{dG_\theta(\boldsymbol{z})}{d\theta} \right], \tag{2}$$

where $s_{\text{data}}$ and $s_{\text{gen}}$ are the score functions that trained of real data and generated data, which point towards the higher density of data for $p_{\text{gen},t}$ and $p_{\text{data},t}$. The noisy sample $\boldsymbol{x}_t$ is obtained by the diffusion forwarding process $\boldsymbol{x}_t = \alpha_t \hat{\boldsymbol{x}}_0 + \sigma_t \epsilon, \ \epsilon \sim \mathcal{N}(0, I)$, where $\hat{\boldsymbol{x}}_0$ is the generated output of the few-step student model $G_\theta$. $\alpha_t, \sigma_T > 0$ are defined by the noise schedule. $\boldsymbol{z} \sim \mathcal{N}(\boldsymbol{0}, \boldsymbol{I})$ is the Gaussian distribution.

The true data score $s_{\text{data}}$ is estimated by the original teacher model, while the generation score $s_{\text{gen}}$ is estimated by a "teaching assistant" (TA) model, which is a fine-tuned version of the teacher. The TA model and the few-step generator are typically trained in an alternating fashion. The TA model is fine-tuned on the standard diffusion denoising loss on the generated data from the student model:

$$\mathcal{L}_{\text{gen}}^\phi = \left\| \epsilon_{\text{gen}}^\phi(\boldsymbol{x}_t, t) - \epsilon \right\|_2^2. \tag{3}$$

## 3.2 ADVERSARIAL SELF-DISTILLATION

A key drawback of DMD is its lack of flexibility. The resulting student model is optimized for a single, fixed number of steps, requiring a separate distillation for each desired configuration (e.g., 4 steps vs. 2 steps). To overcome this rigidity, we introduce a method that aligns the outputs of a single few-step model across varying step counts.

Our core idea is to align the $n$-step denoising distribution with the $(n+1)$-step version. This ensures that the model can maintain consistent, high-quality output regardless of the chosen few-step setting. Since adversarial distillation objectives (Sauer et al., 2024b; Lin et al., 2024) are known to preserve sharpness and fine details better than DMD under low-step constraints, this alignment is performed using a discriminator $D_\psi^n$. This discriminator's task is to make the outputs of adjacent denoising steps indistinguishable through a relativistic pairing GAN objective (Jolicoeur-Martineau, 2018). The adversarial self-distillation loss is represented as:

$$\mathcal{L}_{\text{ASD}}(\theta, \psi) = \underset{\substack{\boldsymbol{z_1} \sim \mathcal{N}(\boldsymbol{0}, \boldsymbol{I}) \\ \boldsymbol{z_2} \sim \mathcal{N}(\boldsymbol{0}, \boldsymbol{I})}}{\mathbb{E}} \left[ f \left( D_\psi^n \left( \Psi \left( G_\theta^n(\boldsymbol{z_1}) \right) \right) - D_\psi^n(\Psi \left( G_\theta^{n+1}(\boldsymbol{z_2}) \right)) \right) \right], \tag{4}$$

where $G_\theta^n$ tries to maxmize $\mathcal{L}_{\text{ASD}}$ and $D_\psi^n$ is optimizied to minimize it. $f(t) = -\log \left( 1 + e^{-t} \right)$ is drawn from classic GAN (Goodfellow et al., 2020; Nowozin et al., 2016) and $\Psi$ represents the adding

---

**Algorithm 2** Inference Procedure with First-Frame Enhancement

---

**Require:** Intensive denoising timesteps $\{t_1, \ldots, t_T\}$, reduced denosing timesteps $\{t_1, \ldots, t_R\}$, video length $L$, few-step autoregressive video generator $G_\theta$,

1: **for** $i = 1$ to $L$ **do**
2:     **Initialize:** $x^i \sim \mathcal{N}(0, I)$
3:     $K = \begin{cases} T & \text{if } i = 1 \\ R & \text{otherwise} \end{cases}$
4:     **for** $j = 1$ to $K$ **do**
5:         **Predict** $\hat{x}_0^i = G_\theta(x^i, t_j)$
6:         **Update** $x^i = \Psi(\hat{x}_0^i, \epsilon, t_{j+1})$, where $\epsilon \sim \mathcal{N}(0, I)$
7:     **end for**
8: **end for**
9: **Return** $\{\hat{x}_0^i\}_{i=1}^L$

---

noise process. $G_\theta^n$ and $G_\theta^{n+1}$ are the generated output of the few-step generator with $n$ steps and $(n+1)$ steps.

The few-step generator is trained with both DMD loss and ASD loss:

$$\mathcal{L}_{\text{total}} = \mathcal{L}_{\text{DMD}} + \alpha * \mathcal{L}_{\text{ASD}}, \tag{5}$$

where $\alpha$ is the hyperparameter that weights the two loss functions. The detailed distillation process is shown in Alg. 1.

Crucially, aligning the outputs of two adjacent few-step predictions is a much more stable training objective than directly aligning a few-step model with its multi-step teacher. The smaller distributional gap between adjacent steps makes our few-step generator significantly easier to train. This approach also provides a unique benefit: the $n$-step generator receives a supervision signal not only from the teacher model's trajectory but also from the $(n+1)$-step prediction, a novel form of self-distillation that further boosts the model's generation quality.

### 3.3 FIRST-FRAME ENHANCEMENT

Existing causal video generation methods typically employ a uniform number of denoising steps for each frame. However, in causal generation process, the quality of subsequent frames is critically dependent on the preceding frames. The quality defects in earlier frames are highly likely to be propagated throughout the rest of the sequence. This is particularly true for the first frame, which has no prior context and must synthesize a high-quality initial state from a zero-data starting point. Consequently, the generation of the first frame requires dedicated attention to mitigate accumulated

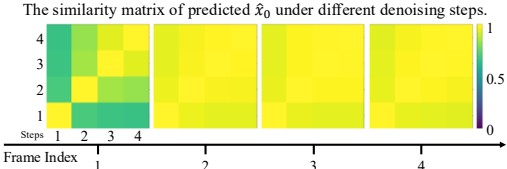

The similarity matrix of predicted $\hat{x}_0$ under different denoising steps.

Figure 4: Cosine similarity matrices of different frames in causal diffusion video generation. Each matrix shows the similarity of the predicted $\hat{x}_0$ between different denoise steps (from 1 to 4).

error and ensure high overall video quality. Fig. 4 shows the similarity of predicted $\hat{x}_0$ for different video frames across various denoising steps. It reveals that the first frame has low similarity across steps, indicating that each denoising step is crucial. In contrast, subsequent frames exhibit higher similarity, suggesting a greater redundancy that makes them more suitable for a few-step prediction. These results are based on the Self Forcing model and are consistent across multiple prompt variants and random seeds. We include additional results across scenarios in the Appendix Section C.

Based on this observation, we propose a novel First-Frame Enhancement (FFE) denoising strategy, as detailed in Alg. 2. The first frame undergoes a more intensive denoising process, requiring a minimum of four steps, while subsequent frames can be generated with a significantly reduced number of steps, such as one or two. This frame-based control allows us to enhance the quality of the generated video in the few-step scenario.

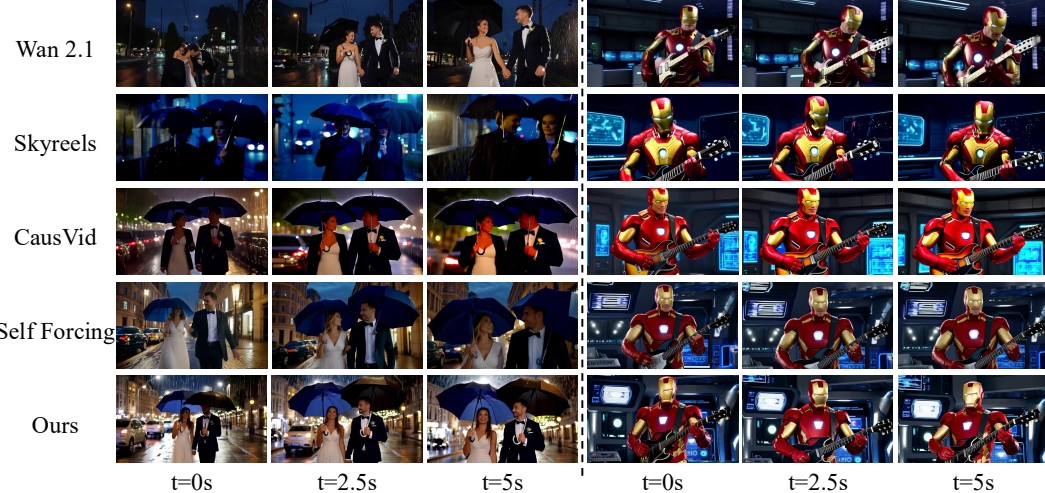

Figure 5: Qualitative comparisons. We visualize videos generated by Ours against those by Wan2.1 (Wang et al., 2025), SkyReels-V2 (Chen et al., 2025), CausVid (Yin et al., 2025) and Self Forcing (Huang et al., 2025) at 4-step generation. All models share the same architecture with 1.3B parameters.

## 4 EXPERIMENTS

### 4.1 IMPLEMENTATIONS

Our approach employs a causal video distillation framework based on the Self Forcing (Huang et al., 2025) training paradigm. The causal model architecture builds upon the Wan2.1-T2V-1.3B (Wang et al., 2025) backbone, a Flow Matching based model (Lipman et al., 2023). We adopt CausVid's (Yin et al., 2025) initialization protocol to stabilize early causal training phases via asymmetric distillation from a pre-trained bidirectional teacher model. For training data, we utilize the exact text prompts from Self Forcing for fair comparison. The diffusion process is optimized with a 4-step denoising schedule during training, while using our step-skipping strategy during inference to accelerate generation. Adversarial training objective is incorporated via integration of RpGAN (Jolicoeur-Martineau, 2019) objectives with R1 (Mescheder et al., 2018) and R2 regularization terms following R3GAN (Huang et al., 2024a). And we use the frozen fake score function as the backbone of the discriminator following DMD2 Yin et al. (2024a). To reduce computational cost and the number of parameters, we model discriminators $D^n$'s output logits as the n-th dimensional output logit of the final layer, thus different $D^n$ share the same backbone and classifier head. The performance is rigorously assessed using VBench (Huang et al., 2024b) for multidimensional evaluation and user preference studies to quantify human-perceived visual quality and semantic alignment.

### 4.2 COMPARISON WITH EXISTING BASELINES

Following the protocol of Self-Forcing, we compare with representative open-source text-to-video models under various inference steps, ranging from 64 to 4, ensuring a comprehensive assessment. For fair comparison under 2-step and 1-step generation, we additionally train 2-step and 1-step distilled versions of Self-Forcing and evaluate them against our model. As shown in Tab. 1, our model achieves slightly better performance compared to Self-Forcing under 4-step generation. Notably, in both the 2-step and 1-step settings, our approach outperforms the specifically distilled versions of Self-Forcing without requiring additional

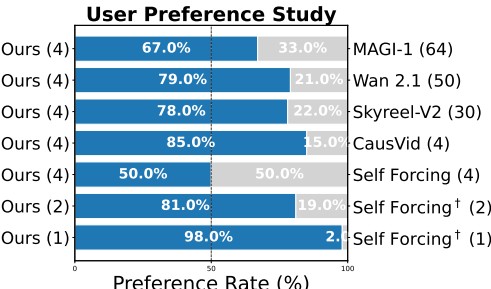

Figure 6: User preference study. "(n)" = number of denoising steps. † denotes the retrained version under 1 & 2 step settings.

Table 1: Comparison with relevant baselines. We compare our method with representative open-source video generation models of similar parameter sizes and resolutions. The n-step generation with FFE strategy is denoted with $n^*$. $\dagger$ denotes the retrained version under 1 & 2 step settings.

| Model | #Params | Resolution | Denoising Steps↓ | Evaluation scores ↑ | | |
| --- | --- | --- | --- | --- | --- | --- |
| | | | | Total Score | Quality Score | Semantic Score |
| *Many Steps* | | | | | | |
| MAGI-1 (Sand-AI, 2025) | 4.5B | 832×480 | 64 | 79.18 | 82.04 | 67.74 |
| Wan2.1 (Wang et al., 2025) | 1.3B | 832×480 | 50 | **84.26** | **85.30** | **80.09** |
| SkyReels-V2 (Chen et al., 2025) | 1.3B | 960×540 | 30 | 82.67 | 84.70 | 74.53 |
| NOVA (Deng et al., 2025) | 0.6B | 768×480 | 25 | 80.12 | 80.39 | 79.05 |
| LTX-Video (HaCohen et al., 2024) | 1.9B | 768×512 | 20 | 80.00 | 82.30 | 70.79 |
| Pyramid Flow (Jin et al., 2025) | 2B | 640×384 | 20 | 81.72 | 84.74 | 69.62 |
| *4 Steps* | | | | | | |
| CausVid (Yin et al., 2025) | 1.3B | 832×480 | 4 | 81.20 | 84.05 | 69.80 |
| Self Forcing (Huang et al., 2025) | 1.3B | 832×480 | 4 | 84.31 | 85.07 | **81.28** |
| Ours | 1.3B | 832×480 | 4 | **84.38** | **85.16** | 81.25 |
| *2 Steps* | | | | | | |
| Self Forcing$^\dagger$ (Huang et al., 2025) | 1.3B | 832×480 | 2 | 83.49 | 84.20 | 80.62 |
| Ours | 1.3B | 832×480 | $2^*$ | **84.32** | **85.15** | **81.02** |
| *1 Steps* | | | | | | |
| Self Forcing$^\dagger$ (Huang et al., 2025) | 1.3B | 832×480 | 1 | 80.62 | 81.19 | 78.35 |
| Ours | 1.3B | 832×480 | $1^*$ | **83.89** | **84.55** | **81.24** |

parameter optimization. For instance, under 1-step inference, our model exceeds Self-Forcing by 3.27 points in Total Score.

As illustrated in Fig. 5, our method is capable of generating high-fidelity videos using only approximately 8% and 13% of the denoising steps required by Wan 2.1(Wang et al., 2025) and SkyReels(Chen et al., 2025), respectively, while achieving visually superior results. This leads to a substantial reduction in computational cost. Moreover, under the same number of denoising steps, our approach yields improved detail and visual quality compared to CausVid(Yin et al., 2025) and better video details compared to Self-forcing(Huang et al., 2025). Fig. 6 shows the user study results comparing our model against several important baselines. Our approach is consistently preferred over many-step baselines, including the many-step Wan2.1 that our model is initialized from. Meanwhile, our results are 70% better than CausVid (Yin et al., 2025) and on par with Self Forcing, since both our model and Self Forcing use identical DMD-based supervision in the 4-step setting. For the extreme 1-step and 2-step generation setting, our method yields a significantly 96% and 62% preference over Self-Forcing, respectively. This demonstrates the effectiveness of our proposed ASD and FFE strategy in improving extremely few-step video generation.

### 4.3 ABLATION STUDY

To better understand the contribution of each component to skip-step generation, we conducted a comprehensive ablation study, as shown in Tab. 2. As shown in the first two rows, using identical training data, our ASD training method consistently improves the quality of both 1-step and 2-step generation during inference. For example, it raises the Total Score by 2.52 and the Semantic Score by 6.84 under one-step generation. Comparison between the first and third rows reveals that the FFE inference strategy leads to substantial gains in overall video quality—notably, a 4.19 increase in Total Score and a 10.64 improvement in Semantic Score for one-step generation, even exceeding the quality of the original two-step generation results (row 1). Moreover, combining both methodologies further enhances performance, as demonstrated in the last row.

We qualitatively compare the quality of generated outputs with and without our proposed ASD training and FFE inference strategies, as illustrated in Fig. 7. For simplicity, n-step generation with the FFE inference strategy is denoted as $n^*$-step. When adopting the ASD training strategy, the generation quality at $2^*$-step is comparable to that achieved with 4 steps, while even under the

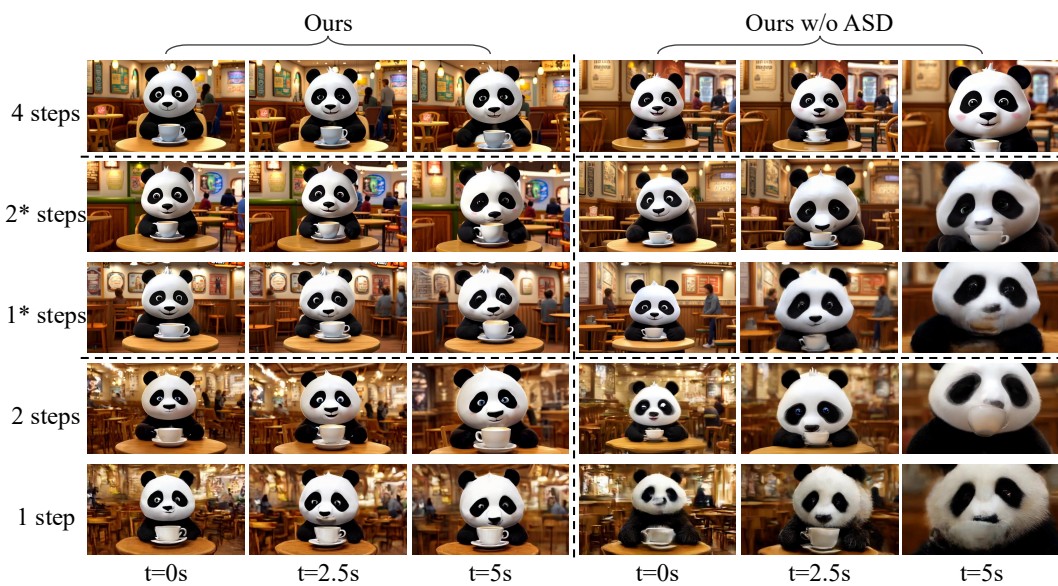

Figure 7: Qualitative comparison illustrating the effects of training and inference strategies on video generation. Left: results using the ASD training method; right: results without it. For simplicity, inference with the FFE strategy is denoted with $*$.

Table 2: Ablation Study. The symbols ✓ and × denote the inclusion and exclusion of the corresponding strategy, respectively. ASD refers to the Adversarial Self-Distillation strategy in Section 3.2, and FFE denotes the First-Frame Enhancement strategy introduced in Section 3.3.

| Methods | | 2-step Generation ↑ | | | 1-step Generation ↑ | | |
|---|---|---|---|---|---|---|---|
| ASD | FFE | Total Score | Quality Score | Semantic Score | Total Score | Quality Score | Semantic Score |
| × | × | 82.61 | 83.84 | 77.68 | 78.13 | 80.33 | 69.31 |
| ✓ | × | 83.28 | 84.17 | 79.69 | 80.65 | 81.77 | 76.15 |
| × | ✓ | 83.80 | 84.65 | 80.40 | 83.04 | 83.81 | 79.95 |
| ✓ | ✓ | **84.32** | **85.15** | **81.02** | **83.89** | **84.55** | **81.24** |

challenging $1^*$-step setting, video quality is well maintained. Moreover, at both 2-step and 1-step inference, the frame quality is noticeably superior to variants trained without the ASD strategy. This shows the effectiveness of our ASD training strategy under skip-step generation. More specifically, the variant trained without the ASD strategy exhibits obvious error accumulation under the $2^*$ setting, resulting in noticeable background shifts and character blurring at t = 5s. And under the $1^*$-step condition, the severe blurring occurs as early as t = 2.5s. As demonstrated by the comparison between rows 2 & 4 and rows 3 & 5, the FFE inference strategy that increases the denoising steps for the initial frame substantially enhances overall video quality by effectively mitigating artifacts such as background and subject blurring in the first frame.

## 5 CONCLUSION

In this work, we introduce an Adversarial Self-Distillation training objective for causal video diffusion models' distillation, along with a First-Frame Enhancement inference strategy for efficient sampling. The adversarial objective encourages the $n$-step generated video distribution to approximate that of the $(n+1)$-step generation, thereby progressively improving few-step generation quality. During inference, we explicitly distinguish between the first frame and subsequent frames in causal video generation, allocating different numbers of denoising steps to enhance overall video quality with low computational overhead. Experiments demonstrate that our distilled model outperforms other baselines under 1- and 2-step generation configurations.

## 6 REPRODUCIBILITY STATEMENT

A comprehensive description of the implementation is provided in Section 4.1 and Section A. The source code and additional comparison videos are included in the supplementary material and will be released in the future.

## 7 ETHICS STATEMENT

This work focuses on improving the efficiency of diffusion-based video generation through distillation. Our contributions are methodological and technical in nature, and do not involve human subjects, personal or sensitive data, or medical applications. The experiments are conducted on publicly available benchmark datasets (e.g., VBench), which are widely used in prior research and do not contain personally identifiable information.

We confirm that this research adheres to the ICLR Code of Ethics and complies with all relevant institutional and legal standards. No conflicts of interest or external sponsorship influencing the work are present.

## 8 ACKNOWLEDGEMENTS

This work was supported by the National Natural Science Foundation of China under grant 62372341. The authors were partially supported by the Intelligent Computing Center of the National Cybersecurity Talent and Innovation Base, Wuhan.

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

## A    DETAILS OF IMPLEMENTATIONS

Our implementation is largely based on the code from Self-Forcing(Huang et al., 2025). Specifically, we adopt the chunk-wise DMD (Yin et al., 2024b) training variant of Self-Forcing, with all training conducted in the VAE latent space using a chunk size of 3. Following Self-Forcing, we utilize the Wan 2.1 14B model as the real score estimator. The model is trained with a batch size of 8 for 3,000 steps.

For Adversarial Self-Distillation, we incorporate the R1 and R2 regularizations (Mescheder et al., 2018) from the R3GAN (Huang et al., 2024a) objective.

$$\mathcal{L}_{\text{reg}} = \frac{1}{2}\mathbb{E}_{t,x_t^{n+1},x_t^n,\epsilon,\hat{\epsilon}} \left[\|f_\psi(x_t^{n+1}) - f_\psi(x_t^{n+1} + \sigma \cdot \epsilon)\|_2^2 + \|f_\psi(x_t^n) - f_\psi(x_t^n + \sigma \cdot \hat{\epsilon})\|_2^2\right] \quad (6)$$

$$\mathcal{L}_D(\psi) = -\mathbb{E}_{t,x_t^{n+1},x_t^n} \left[\log\left(\text{sigmoid}\left(f_\psi(x_t^{n+1}) - f_\psi(x_t^n)\right)\right)\right] + \lambda\mathcal{L}_{\text{reg}} \quad (7)$$

$$\mathcal{L}_G(\theta) = -\mathbb{E}_{t,x_t^{n+1},x_t^n} \left[\log\left(\text{sigmoid}\left(f_\psi(x_t^n) - f_\psi(x_t^{n+1})\right)\right)\right] \quad (8)$$

where $x_t^{n+1} \sim p_{\theta,n+1,t}$, $x_t^n \sim p_{\theta,n,t}$ are the noisy (n+1)-step generation data and n-step generation data, respectively, $\epsilon$ and $\hat{\epsilon}$ are Gaussian noise sampled from $\mathcal{N}(0,1)$, and $f_\psi$ is the critic network (discriminator) of GAN. We use $\lambda = 600$, $\sigma = 0.05$ for all experiments. The generator, fake score estimator, and discriminator are optimized alternately with a ratio of 1:4:1, following(Yin et al., 2024a; Huang et al., 2025). Following Self-Forcing, we inserted additional cross-attention layers and classification heads (serving as discriminator heads) at layers 12, 21, and 29 of the fake score model. These discriminator heads operate solely on backbone features and query tokens, without access to the timestep t. This implementation detail will be explicitly included in the revised manuscript.

Note that we discard ASD Loss for the last step denoising. For simplicity, we reserve the index n of the n-th discriminator $D_n$ that discriminates the $n$-step generation distribution from that of $(n+1)$-step here.

For the retrained Self-Forcing baselines, we use exactly the same hyperparameters as the official implementation, except for changing the student model's number of denoising step list from [1000, 750, 500, 250] to [1000, 500] or [1000]. Specifically, we used: real score CFG weight: 3.0, optimizers: AdamW for both generator and discriminator with $\beta_1 = 0$, $\beta_2 = 0.999$, $\epsilon$ = 1e-8, weight decay = 0.01, learning rate (generator): $2e-6$, learning rate (discriminator): $4e-7$, generator/discriminator update ratio: 5:1. Training is monitored until convergence.

## B    VBench Scores Across All Dimensions

In Figure 8, we evaluate our method and Self-Forcing across all 16 VBench metrics under both 1-step and 2-step generation settings. Our method consistently outperforms Self Forcing, achieving notably higher scores in semantic alignment-particularly in object class, multiple objects, spatial relationships, and scene. Moreover, it demonstrates superior performance in dynamic modeling, as evidenced by a significantly higher dynamic degree score.

## C    Causal Denoising Trajectory Similarity Across Scenarios

To further validate the generality of the phenomenon in Section 3.3, we apply the analysis to both our method and Self-Forcing across three diverse scenarios: VBench(Huang et al., 2024b), complex object motion, and high camera motion. For each, we record denoising trajectories from 200 video samples, yielding Figure 9. The figure shows that, across all scenarios, the first frame exhibits low similarity across denoising steps, indicating each step is critical. In contrast, subsequent frames show high inter-step similarity, suggesting greater redundancy and thus greater suitability for few-step prediction. Moreover, our method achieves even higher similarity in later frames, demonstrating that ASD enables the student to match multi-step quality with fewer steps, effectively supporting step-skipping inference.

## D    Reduced Distributional Gaps Between Adjacent Denoising Steps

To empirically validate the smoother transitions between adjacent denoising steps, we conduct a quantitative analysis using the Fréchet Video Distance (FVD). Table 3 presents FVD values computed between outputs at step $n$ and step $n + 1$ (i.e., adjacent steps), as well as between step $n$ and the 50-step 14B teacher model.

As illustrated in Table 3, the FVD between adjacent denoising steps is consistently lower than that between each step and the final teacher output. For instance, at step 1, the adjacent-step FVD is

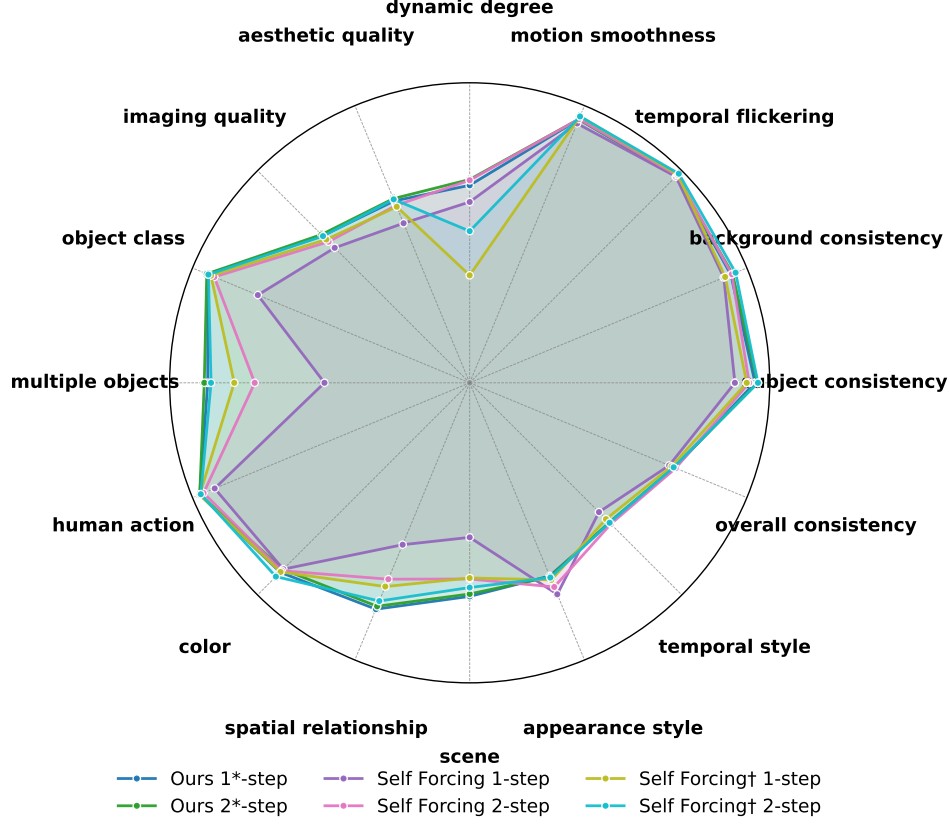

Figure 8: VBench scores visualization. We compare our results with Self Forcing Huang et al. (2025) variants using all 16 VBench metrics under 1-/2-step setting. The retrained 1-/2-step version denotes as Self Forcing$^{\dagger}$.

Table 3: Fréchet Video Distance (FVD) between video distributions at different denoising steps. Lower FVD values indicate smaller distributional gaps.

| Comparison | Step 1 | Step 2 | Step 3 | Step 4 |
|---|---|---|---|---|
| $n \rightarrow n+1$ | 732 | 1136 | 441 | N/A |
| $n \rightarrow$ teacher | 1836 | 1646 | 1454 | 1448 |

732, much smaller than 1836 for the teacher-step comparison. These results provide quantitative support that the distributional discrepancy between consecutive steps is substantially smaller than the divergence from the teacher distribution. The Adjacent-Step Distribution (ASD) objective capitalizes on this property by aligning each step with its immediate successor, which exhibits lower distributional divergence. Overall, these findings corroborate the core assumption of our approach: transitions between adjacent denoising steps are indeed smoother.

## E  FEW-STEP LONG-VIDEO GENERATION EVALUATION

To demonstrate that our method enables high-quality, long-video generation with few (1 or 2) denoising steps—and effectively mitigates error accumulation—we generate 20-second continuations using both 1-step and 2-step inference for our approach and Self-Forcing, and evaluate them on VBench(Huang et al., 2024b).

As shown in Table 4, our method overall outperforms Self-Forcing both quantitatively and qualitatively in long-video generation. For visual comparison, Figure 10 presents side-by-side 2*-step (ours) vs. 2-step (Self-Forcing) 20-second samples. Our results exhibit clearly superior performance in motion

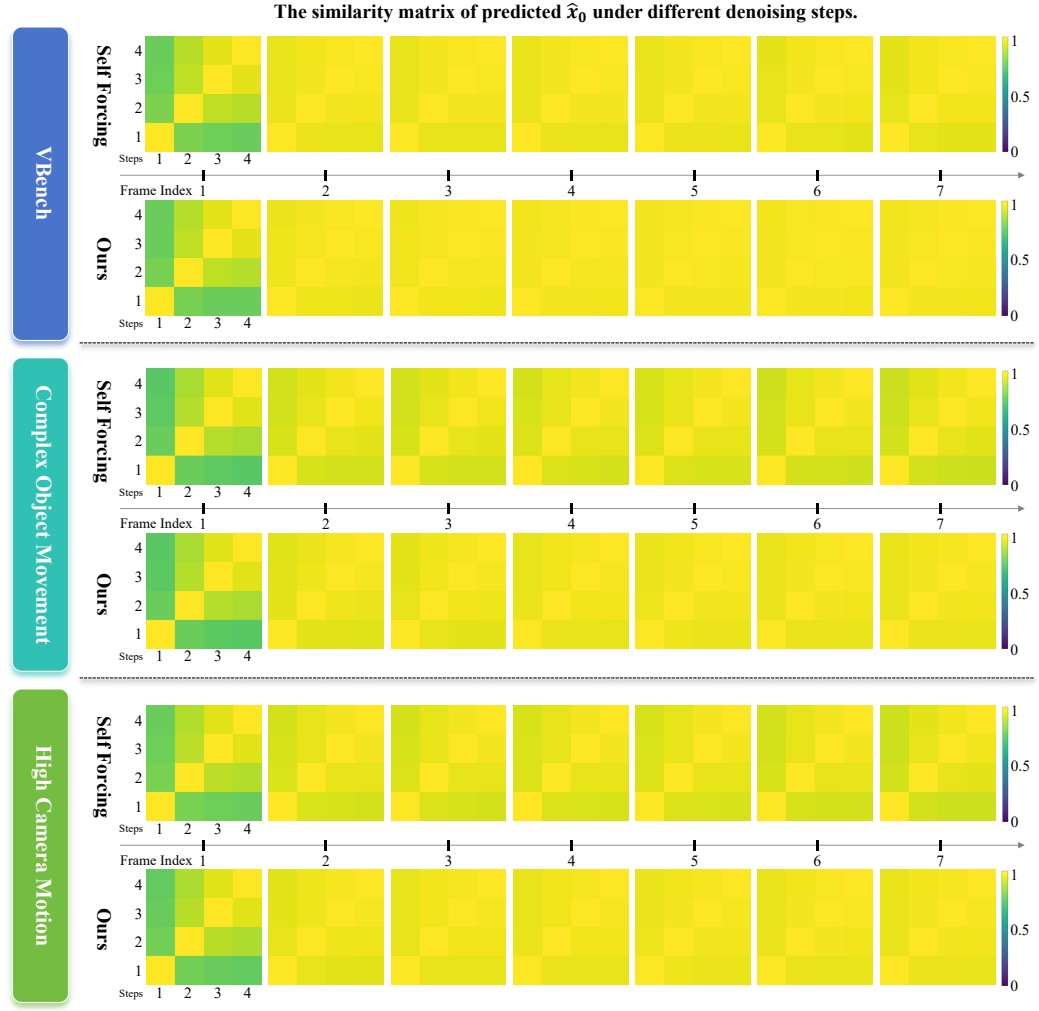

Figure 9: Cosine similarity matrices of predicted $\hat{x}_0$ across denoising steps (1 to 4) in causal diffusion video generation, comparing our method and Self-Forcing (Huang et al., 2025). We report results averaged separately for VBench, complex object motion, and high camera motion.

dynamics (rows 3, 4, 6), visual detail and fidelity (rows 1, 2, 3, 6), and camera motion realism (rows 1, 2, 4).

These results confirm that ASD not only reduces computational cost but also enhances temporal coherence and visual quality in extended video generation.

Table 4: VBench evaluation scores for our method vs. Self-Forcing under few-step long-video (20-second) inference settings

| Method | Total Score | Quality Score | Semantic Score |
|---|---|---|---|
| Self Forcing(Huang et al., 2025) (2-step) | 0.8250 | 0.8293 | 0.8076 |
| Ours (2*-step) | 0.8263 | 0.8329 | 0.7998 |
| Self Forcing(Huang et al., 2025) (1-step) | 0.8066 | 0.8101 | 0.7923 |
| Ours (1*-step) | 0.8200 | 0.8248 | 0.8011 |

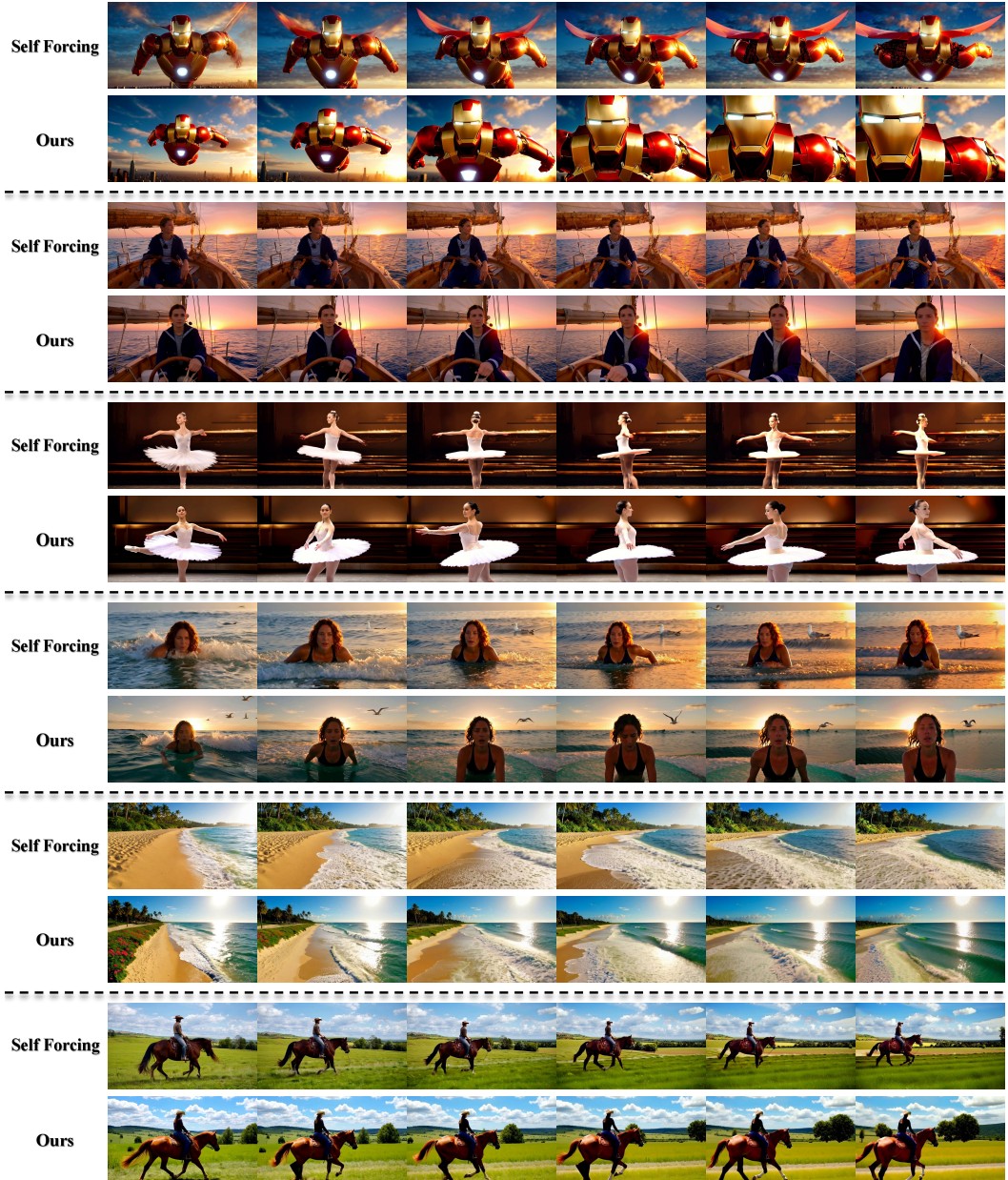

Figure 10: Our method versus Self-Forcing on uniformly sampled frames (including first and last) from 20-second long videos.

## F HYPERPARAMETER ANALYSIS

To investigate the impact of the ASD loss on model performance, we evaluated different variants of the hyperparameter $\alpha$ in the Equation (5) and computed the corresponding Total Score on VBench under three inference settings, as illustrated in Figure 11. The results indicate that at 4-step generation, the Total Scores across different model variants are comparable. When using $2^*$-step generation, the variant without ASD Loss ($\alpha = 0$) exhibits a noticeable performance drop, while other variants maintain performance levels similar to those observed in 4-step generation. Under $1^*$-step generation, all methods experience a decline in performance, with the most pronounced degradation occurring when $\alpha$ is set to zero. These findings demonstrate that the proposed ASD training objective enhances video quality under skip generation settings and remains robust across a wide range of hyperparameter

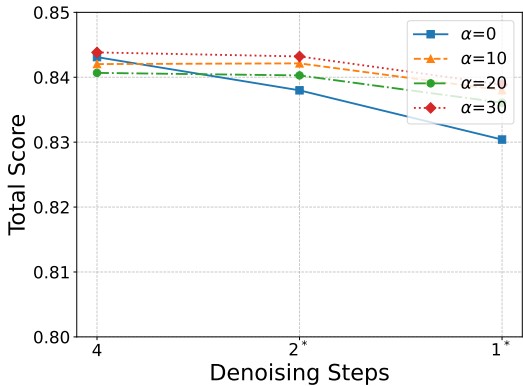

Figure 11: Sensitivity analysis. Impact of parameter $\alpha$ in Equation (5) on the VBench(Huang et al., 2024b) Total Score across three inference scenarios. For simplicity, inference with first-frame enhancement is denoted with $^*$.

values. While absolute values vary slightly across $\alpha$, the trend is consistent: skip-step performance improves markedly whenever $\alpha \neq 0$.

## G    USER STUDY DETAILS

We randomly selected 20 prompts from the VBench(Huang et al., 2024b) prompts expanded by Self-Forcing. Five baseline methods Sand-AI (2025); Huang et al. (2025); Yin et al. (2025); Chen et al. (2025); Wang et al. (2025) were used to generate corresponding videos, which were then paired with corresponding videos generated by our method, resulting in $5 \times 20$ video pairs. During the user study, each participant (a total of 12 participants) evaluated 20 video pairs, including 4 randomly selected pairs from each baseline. The order and pairing of videos were independently randomized for every participant. Participants could not see each other's choices. The final results were averaged across all participants. The user study interface, as shown in Figure 12, displayed a video pair alongside the corresponding prompt and selection buttons. Participants could choose whether the left video was better, the right video was better, or both were of similar quality. To ensure a fair comparison with both one-step and two-step configurations, we retrained the one-step and two-step versions of Self-Forcing.

## H    USAGE OF LLMS

In this work, LLMs were primarily used to assist in grammar checking. All outputs from LLMs were manually reviewed.

## I    TRAINING DYNAMICS ANALYSIS

Figure 13 presents the DMD loss curves of our method and Self-Forcing during training. Notably, our approach exhibits significantly reduced fluctuations in DMD loss compared to Self-Forcing. With ASD, the DMD loss shows lower mean (0.180 vs. 0.198) and dramatically reduced variance (0.00258 vs. 0.01963) compared to pure DMD training, demonstrating that the proposed ASD loss effectively stabilizes distribution matching with the teacher model during training.

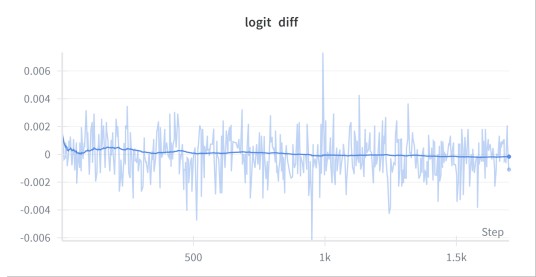

Figure 14: The discriminator's predicted logit difference between n-step and (n+1)-step samples during training.

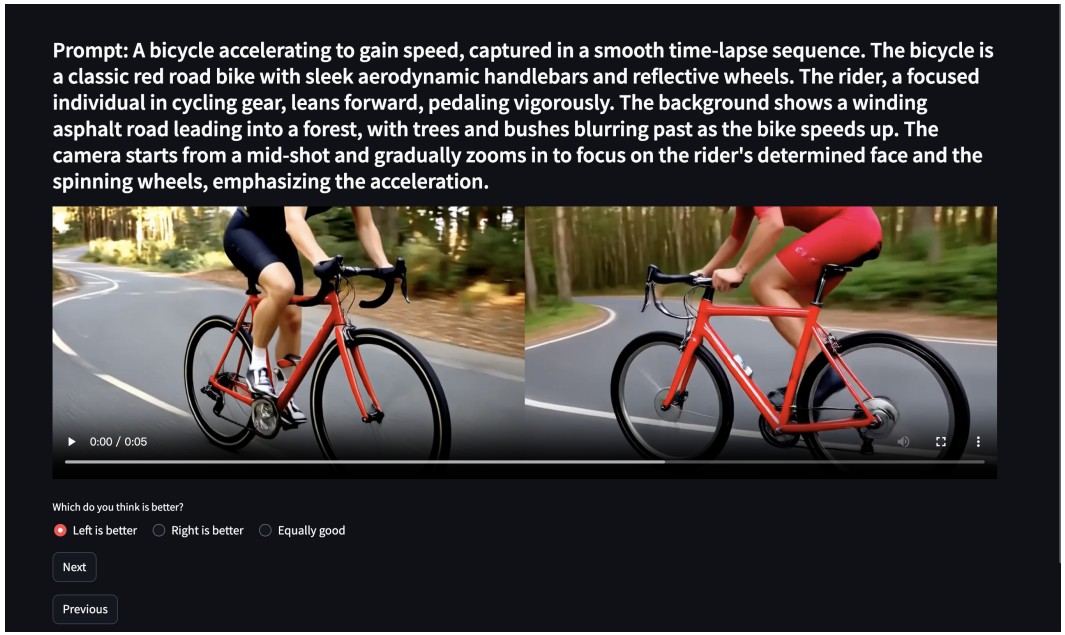

Figure 12: User study interface. Participants were asked to select the higher-quality video in each pair or to indicate if the two were of similar quality.

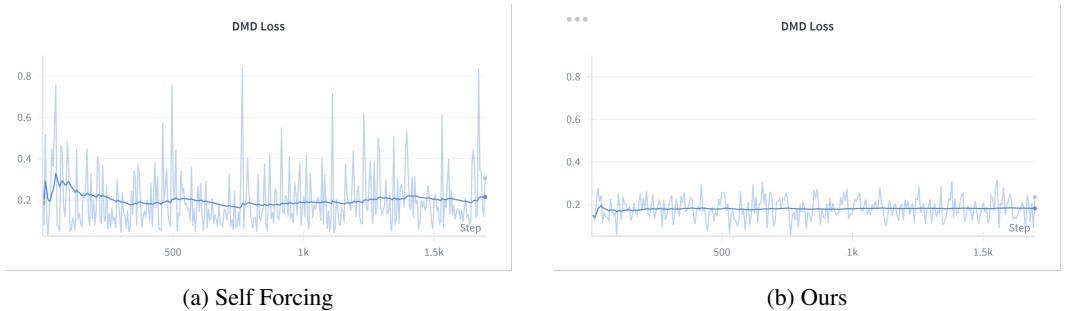

| (a) Self Forcing | (b) Ours |

Figure 13: The DMD loss curves of Self Forcing and our method during training. The DMD loss values remain around 0.2 for both methods, but our method exhibits substantially lower fluctuation.

Figure 14 shows that the discriminator's logit difference between n-step and (n+1)-step samples oscillates calmly around zero, indicating that the generator and discriminator co-evolve stably rather than collapsing or diverging. This behavior supports that ASD training is stable in practice, despite using an adversarial term.

**Training Overhead.** The ASD framework introduces only a lightweight discriminator on top of the existing TA score model's backbone, using shared parameters for all n-step discriminators. This results in minimal extra memory usage compared to traditional DMD. The additional GPU memory required by ASD is +1.3% compared to the Self-Forcing baseline, which is a modest increase. The additional training time is approximately +20% relative to Self-Forcing. Given that ASD allows a single model to handle multiple inference steps (1, 2, 3, 4), this overhead is relatively small compared to the computational cost of training separate models for each step.

