# OpenReview forum: "Towards One-step Causal Video Generation via Adversarial Self-Distillation"
_ICLR.cc/2026/Conference — ICLR 2026 Poster_

### Official Review · Reviewer_nsqN · 2025-10-17

**Soundness:** 3
**Presentation:** 2
**Contribution:** 3
**Rating:** 6
**Confidence:** 4

**Summary:**

The authors propose a method for the distillation of bidirectional video contraption models into autoregressive causal few-steps generators capable of real time inference, basing their work on the popular Self Forcing framework. The paper introduces two advancements with respect to Self Forcing: 1) it enables performing inference with a variable number of sampling steps (1, 2, 4); 2); It improves performance for the case of 2 and 1 step inference with respect to Self Forcing, producing 2 step generation results on par with the original 4 step inference according to VBench scores. The technical contribution enabling these advancements consist of: 1) an adversarial loss term matching N+1 with N step generator distributions; 2) forcing the first frame to be inferred using 4 steps. Both contributions appear sound and easy to implement (authors provide code in the supplement). The quality of the results in convincing as seen qualitatively and in VBench and user studies. Overall, the work is likely to be adopted by the community due to its simplicity, convincing quality of the results and relevance of the problem.

**Strengths:**

- The work tackles the task of making slow bidirectional video models, fast real-time autoregressive generators which if of high practical importance
- The quality of results is convincing as seen in the provided qualitative samples
- Quantitative evaluation confirms qualitative assessment that the method matches or surpasses Self Forcing under the same amount of sampling steps
- The method is simple to implement, and authors provide the source code for review
- Ablation studies show convincingly that ADS and FFE are both having a positive impact on the method and ablations on the optimal value for adversarial loss weighting are shown

**Weaknesses:**

- Tables are missing an analysis of first frame latency and throughput (see Self Forcing). I suggest the authors to report these numbers. FFE will cause first frame latency to match the original 4 steps Self Forcing
- The paper considers only the setting where chunks of 3 latent frames are predicted simultaneously and does not show results for frame-by-frame autoregressive generation. Frame-by-frame prediction is a setting of high practical importance as it minimizes latency. Evaluation would be strengthened by showing qualitatives and quantitative results for this setting too.
- Evaluation is performed on the 5 second setting and no results are shown beyond it. Self Forcing can generalize beyond 5s generation, an important capability for an autoregressive causal generator. Such capability should be demonstrated and evaluated. Without this capability, practical significance of the method would be reduced.
- The paper is unclear in some key parts as Algorithm 1 and discriminator design. See questions.
- Some typos and missing spaces before citations

**Questions:**

- An adversarial term is introduced to match distribution between N+1 and N step predictions, showing performance improvement. Did the authors consider extending use of the same adversarial term to its canonical usage for matching the real data distribution with the 4 step generator distribution similarly to DMDv2?
- Adversarial losses are proposed as the tool for matching the N+1 and N step generator distributions. Why usage of an adversarial term is the ideal choice in this context? Could we have used a DMD formulation instead by introducing additional fake score prediction networks, either one for each value of n, or sharing the same fake score prediction network with conditioning on n? This could result in a more elegant framework only relying on DMD.
- FFE relies on the assumption that generation of the first frame is the hardest, because successive frames can be generated by copying content from the first frame. Thus allocating more sampling steps to the first and less to the subsequent ones makes sense and is shown to improve performance. The assumption however holds less strongly if videos with high camera motion or complex object movements are considered. In this setting each frame will need to generate a more significant amount of content without possibility for copying it from previous frames. Can the authors show that in this setting, their method with 2 steps inference is still matching performance of the original 4 steps Self Forcing?


- Algorithm 1: LL217 shows that a schedule with a fixed number of steps N is instantiated. LL222 suggests that the actual number of sampling steps for the current iteration n is sampled. I believe LL217 should instead instantiate a different schedule for each possible value of n
- Algorithm 1: ll224 suggest a rectified flow setting. I suggest making this explicit in the paper
- Algorithm 1: LL224 LL226 suggest two different time steps are sampled for x^1 and x^2 for use in the adversarial loss term. I'd like to confirm this understanding is correct. Could the authors discuss why this is preferable to having a shared timestep t to ease the role of the discriminator?
- LL352-355 are unclear. How is the discriminator implemented? Does the discriminator receive as input the current timestep t in addition to backbone features? D_n and D^n seem to be used interchangeably
- Eq 2 has incorrect parenthesis
- Could authors report all VBench evaluation metrics in the supplement?
- How is Fig 1 produced? Did the authors perform experimentation on a gaussian mixture?

---

> ### Author Response · Authors · 2025-11-27
> **Response to Reviewer nsqN [1/3]**
>
> **W1: Lack of First frame latency and throughput.**
> In our design, FFE ensures that the first-frame latency matches that of a 4-step denoising process—approximately 1077 ms on our machine—while the throughput for all subsequent frames achieves that of a 1-step generator (9.36 fps).
>
> | Method                     | First Frame Latency | Throughput |
> |----------------------------|---------------------|------------|
> | Self Forcing (4 steps)     | 1077 ms             | 5.40 fps   |
> | Ours (2* steps)            | 1077 ms             | 7.53 fps   |
> | Ours (1* steps)            | 1077 ms             | 9.36 fps   |
>
> ---
>
> **W2: Lack of frame by frame results.**
> We retrained frame-by-frame versions of our model and 2-step Self-Forcing, and evaluated them on VBench. The results show that our method consistently outperforms Self-Forcing—especially in 1-step generation, where our Total Score (0.8076) is significantly higher than Self-Forcing's (0.7554).
>
> | Method                     | Total Score | Quality Score | Semantic Score |
> |----------------------------|-------------|---------------|----------------|
> | Self Forcing (2-step)      | 0.7941      | 0.7936        | 0.7960         |
> | Ours (2*-step)             | **0.8070**  | **0.8084**    | **0.8015**     |
> | Self Forcing (1-step)      | 0.7554      | 0.7640        | 0.7209         |
> | Ours (1*-step)             | **0.8076**  | **0.8090**    | **0.8019**     |
>
> ---
>
> **W3: Lack of evaluation of long video evaluations.**
> We evaluated 20-second videos on VBench and found that our method outperforms Self-Forcing. For qualitative comparison, we provide side-by-side 2*-step (ours) vs. 2-step (Self-Forcing) 20-second video examples in the newly uploaded supplementary material.
>
> | Method                     | Total Score | Quality Score | Semantic Score |
> |----------------------------|-------------|---------------|----------------|
> | Self Forcing (2-step)      | 0.8250      | 0.8293        | **0.8076**     |
> | Ours (2*-step)             | **0.8263**  | **0.8329**    | 0.7998         |
> | Self Forcing (1-step)      | 0.8066      | 0.8101        | 0.7923         |
> | Ours (1*-step)             | **0.8200**  | **0.8248**    | **0.8011**     |
>
> ---
>
> **Q1: Extending the adversarial term to real data.**
> Yes, our method can be directly extended to a full-chain adversarial alignment: 1→2, 2→3, 3→4, 4→real. As shown in the results below, the 2*-step generation quality remains comparable to that of the 4-step baseline, demonstrating that our approach remains effective under this extended real data alignment scheme.
>
> | Method                              | Total Score | Quality Score | Semantic Score |
> |-------------------------------------|-------------|---------------|----------------|
> | Ours with Real Data (4-step)        | 0.8308      | 0.8382        | 0.8008         |
> | Ours with Real Data (2*-step)       | 0.8309      | 0.8385        | 0.8003         |
> | Ours with Real Data (1*-step)       | 0.8251      | 0.8317        | 0.7988         |
>
> ---
>
> **Q2: Can the adversarial loss be replaced by the DMD loss?**
> We appreciate this insightful question. In our experiments and in prior work, we observed:
> ● Pure DMD tends to produce blurry outputs under extreme few-step regimes, especially for 1-step and 2-step generation (e.g., as also shown in prior distillation literature [1,2]). In contrast, adversarial objectives are known to preserve sharpness and fine details better than DMD under low-step constraints (as reported in adversarial diffusion distillation papers [3,4]).
> ● Pure DMD would require training more iterations for additional separate/conditional score models, with an iterative 1:4:4:4:4 optimization schedule in a 4-step setting. This increases the training iterations by approximately 240% compared to the original DMD 1:4 ratio, thereby reducing student optimization efficiency. In contrast, ASD only adds a lightweight discriminator head on top of the existing fake-score estimator (TA model), shared across all n-step discrimination tasks and trained with a simple 1:4:1 schedule. This implementation incurs only a 20% increase in training iterations.
> Thus, the adversarial loss serves as a simple and effective tool to close the small distribution gap between adjacent-step predictions while restoring the sharpness typically lost when using DMD alone.

---

> ### Author Response · Authors · 2025-11-27
> **Response to Reviewer nsqN [2/3]**
>
> **Q3: Does FFE remain effective under dynamic scenes with high camera motion and complex object movement, and can its 2\*-step results match the 4-step performance of Self-Forcing?**
> We collect generation trajectories from 600 videos to compute similarity matrices, with 200 videos each from three datasets: complex object motion, high camera motion, and VBench. The results are presented in the newly uploaded appendix Section C.
> They all exhibit the same trend: the first frame shows low inter-step similarity, while subsequent frames exhibit high similarity. This confirms that first-frame enhancement and step-skipping for later frames generalize across diverse video content. Interestingly, our method achieves obviously higher denoising similarity in subsequent frames than Self-Forcing, demonstrating that ASD enables the student to match multi-step quality with fewer steps.
>
> In addition, as shown in the table below, compared to 4-step Self-Forcing, our fewer-step results achieve higher motion dynamics (Dynamic Degree), greater scene complexity (Multiple Objects), and comparable visual fidelity (Aesthetic Quality, Imaging Quality).
> We have also included additional video comparisons on challenging scenarios in the uploaded supplementary.
>
> | Method                     | Aesthetic Quality | Imaging Quality | Dynamic Degree | Multiple Objects |
> |----------------------------|-------------------|-----------------|----------------|------------------|
> | Self forcing（4-step）     | 0.6581            | 0.6936          | 0.6416         | 0.8545           |
> |                            |                   |                 |                |                  |
> | Self forcing (2-step)      | 0.6618            | 0.6914          | 0.5055         | 0.8620           |
> | ours （2*-step）           | **0.6645**        | **0.6988**      | **0.6777**     | **0.8846**       |
> |                            |                   |                 |                |                  |
> | Self forcing (1-step)      | 0.6350            | 0.6757          | 0.3583         | 0.7847           |
> | ours （1*-step）           | **0.6562**            | **0.6932**          | **0.6583**     | **0.8730**       |
>
> ---
>
> **Q4：Alg.1: Should a separate time schedule be instantiated for each possible value of n?**
> In our training setup, we employ a fixed time schedule, as indicated at LL217:
> T={250,500,750,1000}, N=4.
> For example:
> ● 4-step generation uses timesteps {1000, 750, 500, 250},
> ● 3-step generation uses {1000, 750, 500},
> ● 2-step uses {1000, 750},
> ● 1-step uses {1000}.
> LL222 states that at each training iteration, we randomly sample a targeted inference step n∈{1,2,3}, and perform ASD by aligning the n-step and (n+1)-step generations.
> We believe the confusion arises because N in LL217 simultaneously denotes:
> 1. the subscript of the predefined time schedule, and
> 2. the maximum number of inference steps used in training.
>
> In contrast, the sampled n in LL222 refers only to the number of generation steps used in this training iteration, not a specific timestep from the schedule.  In addition, in order to share a unified time schedule across different values of n, the 3-step setting does not use evenly spaced timesteps; instead, it directly takes the first three elements from the same time schedule.  We have revised the main paper to make this distinction clearer.
>
> ---
>
> **Q5: Rectified Flow Setting.**
> We thank the reviewer for the suggestion. We have made it explicit in the revision that our implementation follows a flow-based (rectified-flow-style) training formulation.
>
> **Q6: Alg.1: Sampling Timesteps for Adversarial Loss**
> Thank you for catching this. Both noise additions in LL224 and LL226 use the same timestep t. The two samples are aligned under the same noise level, consistent with the DMD formulation.
> We have corrected the text in the revised manuscript.
>
> ---
>
> **Q7: How is the discriminator implemented?**
> Following Self Forcing (adversarial version), we inserted additional cross-attention layers and classification heads (serving as discriminator heads) at layers 12, 21, and 29 of the fake score model. These discriminator heads operate solely on backbone features and query tokens, without additional access to the timestep t. These implementation details have been explicitly included in the revised manuscript.  We have also adopted consistent notation, using D^n throughout the revision.
>
> ---
>
> **Q8: All VBench Evaluation Metrics.**
> Thank you for your suggestion—we have incorporated it into the appendix Section B of the revised manuscript.

---

> ### Author Response · Authors · 2025-11-27
> **Response to Reviewer nsqN [3/3]**
>
> **Q9: Was Figure 1 generated from Gaussian mixture experiments?**
> Figure 1 is intended as a conceptual illustration, highlighting the optimization difference between: direct real vs. fake alignment (DMD), and our adjacent-step alignment (ASD). It is not generated through Gaussian mixture experiments. We have clarified this in the caption to avoid confusion.
>
> ---
>
> **References**
>
> [1] One-step Diffusion with Distribution Matching Distillation. 24cvpr
> [2] Improved Distribution Matching Distillation for Fast Image Synthesis. 24NeurIPS
> [3] Adversarial Diffusion Distillation. arxiv.2311
> [4] Distilling Diffusion Models into Conditional GANs. 24eccv

---

### Official Review · Reviewer_wWot · 2025-10-27

**Soundness:** 2
**Presentation:** 3
**Contribution:** 2
**Rating:** 4
**Confidence:** 4

**Summary:**

This paper addresses the challenges of slow inference speed and error accumulation in causal video generation models. The primary goal is to enable high-quality video synthesis in extremely few denoising steps, the authors propose two main contributions:
1. Adversarial Self-Distillation (ASD): A novel strategy that moves away from traditional distillation. Instead of matching a few-step student model to a multi-step teacher model, ASD adversarially aligns the output distribution of the model's own n-step generation with its (n+1)-step generation.
2. First-Frame Enhancement (FFE): An inference strategy designed to mitigate error propagation. Based on the observation that the initial frame is most critical in causal generation, FFE allocates more denoising steps to the first frame and significantly fewer steps to subsequent ones, enhancing overall video quality with minimal computational overhead.

**Strengths:**

1. Well-Motivated and Significant Problem: The paper addresses a highly relevant and challenging problem in generative AI: achieving high-fidelity video synthesis under extreme computational constraints (one or two inference steps). The motivation is clear, and a successful solution to this problem would have a significant practical impact, making the research direction valuable.
2. Empirically Effective and Intuitive Core Ideas:
a) ASD's Practical Efficacy: The core idea of Adversarial Self-Distillation (ASD)—aligning the model's n-step output with its (n+1)-step counterpart—is an intuitive approach to breaking down a large distillation gap. While its theoretical underpinnings could be further explored, its practical effectiveness is undeniably demonstrated by the experiments. The concept of progressively refining the model based on its own slightly improved outputs proves to be a powerful empirical strategy in the few-step regime.
b) Pragmatic and Data-Driven FFE: The First-Frame Enhancement (FFE) strategy is a pragmatic and effective solution grounded in a clear empirical observation (Figure 4). Although a simple heuristic, it demonstrates a thoughtful consideration of the error propagation dynamics in causal models. This data-driven approach to allocating computational resources where they are most needed is a clever and impactful inference-time optimization.
3. Rigorous and Comprehensive Experimental Validation:
a) Strong and Fair Baseline Construction: The authors' decision to train their own few-step versions of a powerful SOTA model (Self-Forcing) for comparison is a sign of rigorous scientific practice. This "apples-to-apples" comparison effectively isolates the contribution of their proposed methods (ASD and FFE) from confounding variables like model architecture, which makes the reported gains highly credible.
b) State-of-the-Art Empirical Performance: The paper presents compelling quantitative and qualitative results that convincingly demonstrate state-of-the-art performance in the challenging one- and two-step video generation tasks. The significant lead over a fairly-trained baseline, supported by extensive ablation studies (Table 2) and a strong user preference study (Figure 6), provides undeniable proof of the method's empirical superiority.
4. Clarity and High-Quality Presentation: The paper is well-written, clearly structured, and easy to follow. The figures and tables are informative and effectively communicate the core concepts and results, contributing to a high-quality presentation of the work.

**Weaknesses:**

1. Methodological Foundation of ASD Lacks Rigor: The central contribution, Adversarial Self-Distillation (ASD), is built on a foundation that is more intuitive than it is rigorous. The paper's core claims—that the "intra-student gap" is smaller and that adversarial alignment provides "smoother supervision"—are presented as assertions rather than demonstrated principles.
a) There is no formal analysis or empirical measurement to quantify this "gap" (e.g., in terms of a specific distribution divergence metric).
b) The claim of "smoother supervision" from a GAN objective is counter-intuitive, given the well-documented instability of adversarial training. The paper fails to provide evidence to substantiate why this would be the case, especially compared to simpler, more stable alignment objectives.
2. Unaddressed Risk of Error Reinforcement in Self-Distillation: The ASD mechanism, where the model learns from a slightly better version of itself, introduces a significant and unexamined risk of "model drift" or "error reinforcement." If the (n+1)-step generation is flawed (e.g., contains artifacts or mode collapse), ASD could perversely train the n-step model to replicate these very flaws. The paper relies on the DMD loss to anchor the model to the true data distribution but provides no analysis of the training dynamics or the delicate balance required to prevent the self-distillation objective from amplifying its own mistakes.
3. The FFE Strategy is Heuristic and Its Generalizability is Questionable: The First-Frame Enhancement (FFE) strategy is presented as a key contribution, but it is fundamentally an empirically-driven heuristic rather than a principled method.
a) Its justification rests entirely on a single observation on a specific dataset (Figure 4), and its generalizability to different video content (e.g., videos with major mid-sequence scene changes) is not explored.
b) The paper completely ignores the potential negative side-effects of this strategy. Creating a sharp drop in denoising steps between the first and second frames could introduce significant temporal discontinuity and artifacts, undermining the very quality it aims to enhance. This critical aspect is neither analyzed nor discussed.
4. Insufficient Discussion on Training Complexity and Stability: The proposed training framework is remarkably complex, involving a generator, a discriminator, and a "teaching assistant" (TA) score model trained in an alternating fashion. The paper largely overlooks the significant practical challenges this entails. There is no discussion of the training stability, the sensitivity to the delicate balance of multiple loss terms and optimizers, or the total computational overhead of this complex setup compared to simpler distillation baselines. For a paper focused on efficiency, the lack of transparency regarding its own training costs is a notable omission.

**Questions:**

1. The central premise of ASD is that the "intra-student gap" is smaller and easier to bridge than the "teacher-student gap." Could you provide a more formal or empirical justification for this claim? For instance, have you measured this "gap" using any distribution divergence metrics (e.g., KL, Wasserstein) to validate this core assumption?
2. Given the known training instabilities of GANs, the claim that ASD provides "smoother supervision" is counter-intuitive. Could you elaborate on this and provide evidence (e.g., loss curves, gradient norm analysis) to support that the adversarial self-distillation process is indeed more stable than direct distillation from a fixed teacher?
3. The self-distillation mechanism seems to carry an inherent risk of error reinforcement, where the model could amplify its own artifacts over time. How does the framework explicitly guard against this "model drift"? What is the role of the DMD loss in anchoring the training?
4. The training procedure appears significantly more complex than the baseline. Could you provide a more transparent comparison of the training time, computational resources, and overall stability of your method compared to the standard distillation approach used to train the Self-Forcing† baseline?

---

> ### Author Response · Authors · 2025-11-27
> **Response to Reviewer wWot [1/3]**
>
> **W1.1 & Q1: What evidence supports the smoother distributional gaps between adjacent steps?**
> To quantify the distributional gap between generations at different denoising steps, we compute the Fréchet Video Distance (FVD) between: (i) outputs at step n and step n+1, and (ii) outputs at step n and the 50-step teacher.
> As shown in the Table, the FVD between adjacent steps (n vs. n+1) is consistently much smaller than that between step n and the 50-step teacher—for instance, 732 vs. 1836 at 1-step generation.
> This provides direct empirical evidence that adjacent-step distributions are closer, and that the ASD objective leverages a lower-divergence alignment target.
> We have included this analysis and the corresponding table in the revised manuscript.
>
> | FVD           | 1    | 2    | 3   | 4   |
> |---------------|------|------|-----|-----|
> | n → n+1       | 732  | 1136 | 441 | N/A |
> | n → teacher   | 1836 | 1646 | 1454| 1448|
>
> ---
>
> **W1.2 & Q2:**
>
> **(1) Why is adversarial objective adopted to provide "smoother supervision"?**
> *Clarification of "smoother supervision".* We clarify that “smoother supervision” refers not to the adversarial loss, but to the adjacent distribution alignment signal: In addition to DMD loss, aligning n-step with (n+1)-step provides denser and more progressive feedback, serving as a more gradual transition toward the teacher model.
> *The reason for adopting adversarial loss.* Our choice of adversarial supervision is motivated by empirical evidence: Prior work[1,2] shows that DMD alone tends to produce blurry results under extreme few-step settings. In contrast, adversarial distillation performs better in low-step regimes [3,4], preserving sharpness and detail more effectively.
>
> **(2) Is ASD training more stable than direct distillation from a fixed teacher?**
> We monitor the DMD loss for two variants: (1) ASD training and (2) ASD-free training (i.e., direct distillation from a fixed teacher). As shown in the table below or the newly uploaded Appendix Fig.13, ASD yields markedly smoother training: the DMD loss variance drops from 0.01963 (without ASD) to 0.00258 (with ASD). This demonstrates that ASD's step-wise self-supervision effectively stabilizes the distillation process.
> In addition, we monitor the discriminator's output difference between n-step and n+1-step samples during training. As shown in the newly uploaded Appendix Fig. 14, it oscillates stably around zero, with a mean of $-9.23 \times 10^{-5}$. This indicates balanced co-evolution of generator and discriminator without collapse or divergence. This confirms that, in practice, ASD's adversarial term does not compromise training stability.
>
> | DMD Loss      | Mean  | Variance |
> |---------------|-------|----------|
> | ASD-free  | 0.198 | 0.01963  |
> | ASD         | 0.180 | 0.00258  |
>
> ---
>
> **W2 & Q3:**
>
> **(1) What is the role of the DMD loss in anchoring the training?**
> At each denoising step, the model is supervised by both DMD and ASD (as in Equation 5), except at the final step, which relies solely on DMD.
> DMD serves as the primary supervision signal between teacher and student, but the large distributional gap—especially under extreme few-step settings—limits its effectiveness. ASD addresses this by introducing a student-to-student self-supervision signal as an auxiliary loss, which becomes crucial when denoising steps are drastically reduced. Without DMD, distillation cannot proceed; without ASD, optimization under extreme few-step conditions degrades significantly.
> In conclusion, DMD keeps learning from a fixed teacher's true distribution, serving as a final fixed global target that directly aligns the student with the true distribution, rather than an intermediate or surrogate one.

---

> ### Author Response · Authors · 2025-11-27
> **Response to Reviewer wWot [2/3]**
>
> **W2 & Q3 (continue):**
>
> **(2) Does ASD risk drifting from the teacher distribution by relying on self-generated (n+1)-step signals?**
> Below, we explain why ASD does not suffer from model drift or error reinforcement.
> ● Model drift may exists in the following setting (not our setting): Sequentially training separate models for 4-, 3-, 2-, and 1-step inference risks model drift: for instance, training the 2-step model solely under supervision from the 3-step model—without direct guidance from the teacher—can cause it to drift toward the 3-step student rather than the true teacher distribution.
> In contrast, our approach trains all steps jointly, with every student step receiving direct supervision from the fixed teacher (via DMD) while additionally aligning with its next-step counterpart (via ASD) as an auxiliary signal. This dual supervision ensures alignment with the teacher throughout and eliminates the drift problem inherent in sequential distillation.
> ● Empirical evidence of convergence to the teacher, not model drift.
> ASD is always used alongside DMD (Equation 5), where DMD measures the KL divergence between student and teacher.
> Therefore, if a student drifts away from the teacher by ASD would manifest as increased DMD loss and higher variance. However, as shown in the table above or the DMD loss curve in revised Appendix Fig. 13,
> ○ Without ASD (pure DMD), the mean/variance are 0.198/0.01963.
> ○ With ASD, the DMD loss has mean/variance decreases to 0.180 / 0.00258.
> The significantly lower variance and mean under ASD shows that it reduces instability rather than amplifying it, and actually helps the student model stay closer to the teacher distribution throughout training.
> We have included this analysis in the revised paper.
>
> ---
>
> **W3:**
>
> **(1) Does FFE generalize across diverse video content beyond a specific dataset?**
> During rebuttal, we have collected generation trajectories from 600 videos to compute similarity matrices, with 200 videos each from three datasets: complex object motion, high camera motion, and VBench. The results are presented in the newly uploaded appendix Section B. All three datasets exhibit the same trend: the first frame shows low inter-step similarity, while subsequent frames exhibit high similarity. This confirms that first-frame enhancement (FFE) generalizes across diverse video content. To intuitively demonstrate the generalization capability of FFE in such challenging datasets, we have also included additional video comparisons on challenging scenario datasets in the uploaded supplementary.
> The mentioned mid-sequence scene changes (i.e., shot changes) are relatively rare in typical 5-second video generation settings, and even standard Self-Forcing 4-step generation struggles to produce temporally coherent results across such abrupt changes. Consequently, handling shot transitions falls outside the current scope of our optimization, which focuses on improving efficiency and quality under temporally continuous generation.
> However, for future applications such as long video generation with scene changes, a plausible solution—such as adaptively increasing denoising steps at predicted scene transitions—could address this limitation. This, however, requires dedicated modeling and evaluation beyond the scope of our current framework, and we leave it to future work.
>
> **(2) Does a sharp drop in denoising steps introduce temporal artifacts?**
> We find no evidence of obvious temporal artifacts—supported by both quantitative and qualitative results.
> We compute the SSIM between two adjacent frames immediately after FFE (i.e., the first frame generated with 4 denoising steps and the second frame generated with a reduced number of steps). As shown in the table below, reducing from 4 to 2 steps yields nearly identical average SSIM to the consistent 4-step baseline (0.8492 vs. 0.8502; Δ ≈ 0.001). Even with aggressive reduction to 1 step (4 → 1), the average SSIM only decreases slightly to 0.8418 (Δ ≈ 0.0084)—a difference imperceptible to the human eye. Critically, variance remains stable across settings, indicating consistent behavior.
> Qualitatively, sample videos in the appendix confirm the absence of noticeable temporal artifacts under step reduction.
>
> | SSIM          | (4 → 4)-step | (4 → 2)-step | (4 → 1)-step |
> |---------------|--------------|--------------|--------------|
> | Mean          | 0.8502       | 0.8492       | 0.8418       |
> | Variance      | 0.0138       | 0.0134       | 0.0131       |

---

> ### Author Response · Authors · 2025-11-27
> **Response to Reviewer wWot [3/3]**
>
> **W4 & Q4:**
>
> **(1) Discussion of the training stability.**
> Please refer to W1.2&Q2 (2) above.
>
> **(2) The sensitivity to the delicate balance of multiple loss terms and optimizers?**
> We balance DMD loss and ASD loss with α. It is robust since the Hyperparameter Analysis Section in the appendix show that different choices of α consistently yield quality gains under step-skipping inference. For example, when α = 10, the VBench Total Scores for [4, 2*, 1*]-step generation are [0.8420, 0.8421, 0.8379], which are comparable to those at α = 30: [0.8438, 0.8432, 0.8388].
> The discriminators simply share one optimizer, and it worked well.
>
> **(3) What's the computational resources overhead compared to Self Forcing?**
> All ASD discriminators share weights, resulting in minimal overhead—only a 1.3% increase in average GPU memory usage. Due to the additional discriminator training, our method incurs a 20% longer training time compared to Self-Forcing.
> However, our method requires only a single training run to support multiple inference-step configurations, eliminating the need—unlike Self-Forcing—to re-distill a new model for each step setting. Moreover, under the same few-step inference, our approach achieves higher video generation quality.
>
> ---
>
> **References**
>
> [1] One-step Diffusion with Distribution Matching Distillation. 24cvpr
> [2] Improved Distribution Matching Distillation for Fast Image Synthesis. 24NeurIPS
> [3] Adversarial Diffusion Distillation. arxiv.2311
> [4] Distilling Diffusion Models into Conditional GANs. 24eccv

---

### Official Review · Reviewer_MC9Q · 2025-10-27

**Soundness:** 3
**Presentation:** 3
**Contribution:** 2
**Rating:** 6
**Confidence:** 3

**Summary:**

This paper works towards limiting the diffusion denoising steps of hybrid video generation frameworks, that leverage autoregressive models to model temporal dynamics and diffusion-based spatial denoising, to as few as one step. The proposed method falls under the category of model distillation and strongly builds on an existing method called Distribution Matching Distillation with a substantial addition: A novel form of Adversarial Self Distillation is proposed, aligning the student model’s n-step denoising process with its (n+1)-step version on a distribution level. Results on VBench and a custom user study show state of the art results on 1 and 2 step video generation. The model further removes the limitation of fixed inference steps after training, allowing flexibility for multi-step settings.

**Strengths:**

- The paper is generally well written and presented.
- Provided 1 and 2 step video results, both in the paper and supplementary material, show better results than current competitors.
- The method supports both single and few / multi step inference which is a major advantage over fixed step trained methods
- The observation First Frame Strategy seems to be an important observation, by itself already boosting state of the art results
- The influence of ASD and FFE cleanly ablated showing the superior results of the combination

**Weaknesses:**

## Incremental contribution:
- The used components are not fundamentally new in nature. DMD is very well established for model distillation and remains a core component also in this work.
- Similar adversarial diffusion distillation has been proposed before and is well established in the community
- As a such there are no fundamentally new concepts presented, but their combination provides a nice contribution to the current state of research.

## Limited information on experiments:
- There are several information missing on some of the shown experiments
- The user study is missing the number of participants and further statistical values
- For the main comparison for 1 or 2 step distillation, the authors had to retrain Self Forcing. Retraining parameters are missing i.e. indicating that the model has been trained long enough to convergence

## Figure 1:
- Figure 1 seems not really on point, i.e. the adversarial self-destillation on the right seems oversimplified and not aligned with Algorithm 1.

## Minor:
- Algorithm 1, 1: Typo: origianl

**Questions:**

- Eq. 2: isn’t there a bracket missing?
- How is Fig. 4 created? Is this just an example or averaged results? Are the results corresponding to results from Self Forcing or from the provided method?
- Regarding user preference study: Please provide more details on e.g. how many users were selected, were measures taken to ensure independence?
- In the ablation Table 2: Why are the total scores for the first row (ASD, FFE) so much worse than the corresponding pure Self Force values from Table 1?
- The distilled diffusion process is optimized with a 4-step denoising process. Was this number of steps determined to optimal?
- Influence of $\alpha$: Figure 8 already shows influence of alpha on the Total Score. However it is interesting to see that the order of the alpha values with increasing scores (for 2 or 1 step) goes with: alpha=0 < alpha=20 < alpha=10 < alpha=30, which shows a non-monotonuous increase of the total score with increasing alpha. How do you explain this behavior?

---

> ### Author Response · Authors · 2025-11-27
> **Response to Reviewer MC9Q [1/3]**
>
> **W1: What is the key innovation of the proposed method over DMD?**
> We would like to clarify the role of DMD and highlight where our contributions differ fundamentally.
> - **DMD is used as a baseline foundation, not as our novelty.**
>   Recent diffusion distillation methods [1,2,3] commonly rely on DMD or its variants.
>   However, our key contribution is independent of DMD: we introduce a new form of step-wise alignment across the student’s *n*-step and (*n*+1)-step predictions.
>   This type of intra-student, adjacent-step supervision does not exist in DMD and is specifically designed to address extreme few-step scenarios (e.g., 1-step or 2-step generation).
> - **DMD alone degrades significantly under extreme few-step settings, while ASD greatly relieves this degradation.**
>   As shown in Fig. 2 in the original manuscript, Self-Forcing trained purely with DMD shows severe quality drops at 1-step and 2-step inference (blur, missing details). In contrast, ASD provides a more stable and high-fidelity generation even in such extremely low-step settings. In Table 2, ASD boosts the 2-step and 1-step VBench Total Score by 1.23 and 2.52 respectively.
>
> Thus, while DMD serves as an essential component, the core novelty of this work lies in the proposed Adversarial Self-Distillation, which changes how few-step generators are supervised and directly restores performance where DMD-based methods typically fail.
>
> ---
>
> **W2: What is the key innovation of the proposed method over adversarial loss?**
> We would like to clarify the conceptual distinction between prior adversarial distillation methods and our proposed ASD.
> - **Prior adversarial losses are always applied to align generated samples directly with real data, i.e., fake → real alignment.**
>   In contrast, our work is the first to employ the adversarial objective for adjacent-step student model alignment, distinguishing *n*-step from (*n*+1)-step predictions—making the supervision signal qualitatively different from existing adversarial distillation formulations. This enables the distilled model to maintain high generation quality even under step-skipping settings.
> - **The motivation is different from previous adversarial distillation works.**
>   Prior adversarial approaches primarily aim to improve image/video quality by directly matching real-data distributions. In contrast, our goal is to establish step-wise alignment between the student’s own denoising stages, which in turn drives alignment with the real data—enabling high fidelity even at extremely few denoising steps. And the advantage is substantial: our 1-step results already match the 4-step adversarial version Self-Forcing (83.89 vs. 83.88 Total Score).
>
> ---
>
> **W3: Is the method merely a combination of DMD and adversarial loss?**
> We would like to clarify our contribution more precisely.  DMD itself is not our novelty, nor is the use of adversarial loss per se. The core contribution of this work is the adversarial self-distillation strategy explicitly designed for extreme few-step generation, where adjacent-step mutual supervision is crucial and has not been explored in prior literature. After distillation, our model supports adaptive and flexible inference-step configurations to meet the demands of varying computational load application scenarios without retraining.
>
> ---
>
> **W4 & Q3: Details of the user study.**
> Thank you for pointing this out. In total, 12 participants took part in the user study. Each participant evaluated 20 randomly selected video pairs, and:
> - The order and pairing of videos were independently randomized for every participant.
> - Participants could not see each other’s choices.
> - The evaluation interface is shown in Appendix Fig. 12.
>
> We have added these details to the revised manuscript.
>
> ---
>
> **W5: Details of retraining the Self-Forcing.**
> All Self-Forcing baselines were retrained using exactly the same hyperparameters as the official implementation, except for changing the student model’s number of denoising step list from [1000, 750, 500, 250] to [1000, 500] or [1000]. Specifically, we used:
> - Real score CFG weight: 3.0
> - Optimizers: AdamW for both generator and discriminator with β₁ = 0, β₂ = 0.999, ε = 1e-8, weight decay = 0.01
> - Learning rate (generator): 2e−6
> - Learning rate (discriminator): 4e−7
> - Generator/discriminator update ratio: 5:1
>
> Training is monitored until convergence. We have included these hyperparameters in the revision.

---

> ### Author Response · Authors · 2025-11-27
> **Response to Reviewer MC9Q [2/3]**
>
> **W6: Figure 1 seems oversimplified and not aligned with Algorithm 1.**
> Figure 1 is a conceptual illustration highlighting the fundamental difference between: direct real vs. fake alignment (DMD), and our adjacent-step alignment (ASD). For clarity, we omitted the components shared with DMD, focusing only on the conceptual contrast rather than the full training pipeline. To improve accuracy, following your suggestion, we have updated the figure in the revised version to visualize the complete supervision signals, including both the DMD loss and the ASD loss, ensuring consistency with Algorithm 1. Please see the newly uploaded manuscript.
>
> ---
>
> **W7 & Q1: Typo and missing bracket.**
> We thank the reviewer for catching these issues. The typo and equation have been corrected. The mentioned ambiguity or formatting inconsistencies have been fixed in the revised version.
>
> ---
>
> **Q2: More information of Fig. 4?**
> The similarity matrix in Figure 4 is created from one prompt. It uses the first prompt from standard VBench with a Self-Forcing model.
> To provide averaged results, we extensively computed average similarity matrices over 600 prompts, evenly distributed across the three scenarios (200 per scenario), and found that they almost yield the same observation.
> Detailly, we additionally evaluate three scenarios—VBench, high camera motion, and complex object movement—recording average similarity metrics from 200 videos per scenario for both our method and Self-Forcing. All yield statistically consistent results, confirming the robustness of our findings. This analysis has been added to the revised Appendix Section C.
>
> ---
>
> **Q4: Why do the 1-/2-step Total Score in Table 2 (ablation study) show lower scores than those in Table 1 (SOTA comparison)?**
> (1) **Why does Table 2 shows different scores from Table 1?**
> Table 1 includes retrained 1-step and 2-step Self-Forcing models for SOTA comparison, while Table 2 evaluates skip-step inference without specialized retraining for ablation comparison.
> The difference arises from the evaluation protocol and fairness constraints used in each table.
> - Table 1 aims to provide a fair comparison with the best-performing Self-Forcing. Therefore, the Self-Forcing baseline is retrained separately for the 2-step and 1-step settings, following the official procedure for each target inference step.
> - Table 2, on the other hand, is an ablation study designed to isolate the effects of ASD and FFE. To follow the single-variable control principle, all models in Table 2 are trained using the exact same 4-step training configuration, and then evaluated under different inference-step settings.
>
> (2) **Why does Table 2 shows lower scores than Table 1?**
> The lower performance of “Self-Forcing (no ASD, no FFE)” in Table 2 reflects the fact that a 4-step–trained model performs poorly when directly forced to run at 1 or 2 steps. In contrast, our method degrades far less: when reducing from 4-step to 1-step denoising, Self-Forcing’s semantic score drops by 14.97 (84.28 → 69.31), whereas ours declines by only 5.10 (84.25 → 76.15). This demonstrates that our distilled model generalizes better across different steps without retraining.
>
> ---
>
> **Q5: Was the 4-step distillation schedule empirically determined to be optimal?**
> The 4-step distillation schedule is a commonly used setting in distribution-matching-based distillation approaches [4,5]. These methods often use a 4-step schedule due to its effectiveness in balancing training stability and inference quality. As shown in the table, we additionally trained models with 8-step and evaluated their VBench Total Scores under 1-step and 2-step skip-step inference. Overall, the three settings perform similarly.
>
> | Inference Steps | 8-step training | 4-step training |
> |------------------|------------------|------------------|
> | 1 step           | 0.8132           | 0.8065           |
> | 2 step           | 0.8309           | 0.8328           |

---

> ### Author Response · Authors · 2025-11-27
> **Response to Reviewer MC9Q [3/3]**
>
> **Q6: How is the behavior of the ASD loss weight α explained?**
> The observed non-monotonic trend in Figure 8, where the Total Score improves with increasing α but not in a simple linear manner, is influenced by several factors:
> - **Training randomness:**
>   The performance at different α values can be affected by random factors during training, such as the random initialization of model parameters, random seeds, and other stochastic processes. This is evidenced by the different starting points of performance at the 4-step setting of different α values.
> - **Impact of α on skip-step performance:**
>   However, once the model is trained and frozen for inference, the overall trend remains consistent: when α > 0 (with ASD supervision), the performance gap between 1/2-step and 4-step generation is significantly reduced compared to α = 0 (without ASD).
> We have included this explanation in the revision.
>
> ---
>
> **References**
> [1] Self Forcing: Bridging the Train-Test Gap in Autoregressive Video Diffusion. NeurIPS 2025 Spotlight.
> [2] Adversarial Distribution Matching for Diffusion Distillation Towards Efficient Image and Video Synthesis. ICCV 2025 Highlight.
> [3] Learning Few-Step Diffusion Models by Trajectory Distribution Matching. ICCV 2025.
> [4] One-step Diffusion with Distribution Matching Distillation. CVPR 2024.
> [5] Improved Distribution Matching Distillation for Fast Image Synthesis. NeurIPS 2024 Oral.

---

### Official Review · Reviewer_LHmL · 2025-11-01

**Soundness:** 4
**Presentation:** 4
**Contribution:** 4
**Rating:** 8
**Confidence:** 5

**Summary:**

This paper proposes a framework for accelerating causal video diffusion models via Adversarial Self-Distillation (ASD) and First-Frame Enhancement (FFE). The method extends Distribution Matching Distillation (DMD) by introducing a discriminator that aligns the student model’s n-step and (n+1)-step denoising distributions, instead of aligning directly with a multi-step teacher. This step-wise self-alignment aims to stabilize training under extreme few-step (1–2 step) scenarios. Additionally, FFE allocates more denoising steps to the first frame to mitigate error propagation in causal video generation. Experiments on VBench show that the proposed model surpasses Self-Forcing and CausVid under 1-step and 2-step configurations, while achieving comparable performance to multi-step baselines such as Wan2.1 and SkyReels with much fewer steps.

**Strengths:**

1. The paper is clearly written and easy to follow. The proposed methods (ASD and FFE) are well-motivated and clearly elaborated.

2. The results look very impressive. Compared to the Self-Forcing baseline, the 1-step video generation exhibits a great boost in quality. The speed of 1-step causal generation will enable a wider deployment of streaming video generation.

**Weaknesses:**

One minor concern might be conceptual novelty. The main method of this work, ASD, is not fundamentally new [1,2].
Considering the value and impact of 1-step causal video generation, the engineering effort to tune an end-to-end pipeline is a significant contribution, especially that the authors provide the code for replication in the supplementary material.


[1] Zhang et al., SF-V: Single Forward Video Generation Model.

[2] Lin et al., Autoregressive Adversarial Post-Training for Real-Time Interactive Video Generation.

**Questions:**

1. The proposed FFE seems to benefit early generation more, right? Can you provide more results and comparisons for longer generations, such as 10s-20s level?

2. In the user study, the proposed method is on par with Self-Forcing at the 4-step setting (exactly 50%). Can you elaborate more on why ASD is beneficial at 1-2 steps (as in the ablation) but not helpful in the 4-step setting?

---

> ### Author Response · Authors · 2025-11-27
> **Response to Reviewer LHmL [1/1]**
>
> **W1：What distinguishes ASD from the cited method?**
> The cited methods [1,2] both align one-step predictions with large-scale pre-collected data. This entails directly bridging a distributional gap between single-step samples and high-quality samples.
> Our ASD is different: it aligns the student’s *n*-step and (*n*+1)-step predictions, effectively leveraging the student model itself to provide locally consistent supervision—eliminating the need for any additional data collection. After training, our model enables any denoising step generation using one model weight. Our method thus naturally supports adaptive and flexible inference-step configurations to meet the demands of varying computational load application scenarios.
> We have added additional discussion of conceptual differences in the revised paper.
>
> **Q1：Does FFE benefit long video generation quality?**
> We evaluated 20-second videos on VBench and found that our method outperforms Self-Forcing. For qualitative comparison, we provide side-by-side 2\*-step (ours) vs. 2-step (Self-Forcing) 20-second video examples in the supplementary material.
>
> | Method               | Total Score | Quality Score | Semantic Score |
> |----------------------|-------------|---------------|----------------|
> | Self Forcing (2-step) | 0.8250      | 0.8293        | **0.8076**         |
> | Ours (2\*-step)       | **0.8263**      | **0.8329**        | 0.7998         |
> | Self Forcing (1-step) | 0.8066      | 0.8101        | 0.7923         |
> | Ours (1\*-step)       | **0.8200**      | **0.8248**        | **0.8011**         |
>
> **Q2：Why does the proposed method show clear advantages in 1/2-step settings, yet the advantage vanish at 4-step setting?**
> ASD supervision is designed to boost extremely few inference step performance (i.e., 1-/2-step settings) by encouraging the *n*-step's generation to be closer to the (*n*+1)-step's. Therefore, the 1-/2-step generation additionally receives corresponding ASD supervision during training and directly improves performance. However, the final-step (4-step) generation for both our model and Self-Forcing uses identical DMD-only supervision. Therefore, little difference is expected at 4-step inference.
> We have clarified this behavior in the revised manuscript.
>
> ---
>
> **References**
>
> [1] Zhang et al., SF-V: Single Forward Video Generation Model.
> [2] Lin et al., Autoregressive Adversarial Post-Training for Real-Time Interactive Video Generation.

---

### Meta-Review · Area_Chair_jLrd · 2026-01-01

**Summary:**

The paper presents a new method for step distillation of  video diffusion models. The method is based on self-forcing approach with two key differences:
1) Addition of adversarial self distillation, which aligns the student model n-step output with (n+1) step output.
2) First frame enhancement, which simply uses more diffusion steps for the first frame.

The main cluster of undressed concerns stems from reviewer wWot. The paper is indeed have weak methodological foundation. The authors provide some measurements of "intra-student gap", however it is more of an empirical observation. Similarly FFE strategy is heuristic and will not work for videos with frame jumps. Having said that, area chair believe that it is fine to present in a paper a **working** solution, that is motivated by intuition. Given other reviewers positive view of the results presented in the paper, area chair believe this paper is suitable for acceptance.

**Reviewer Concerns:**

- **Comparison for long videos**. Authors provide additional evaluation and results for long videos.
- **Incremental technical contribution**. Area chair believe that results support technical contribution claims. Given the importance of the problem, technical contribution seems sufficient to area chair.
- **Missing some experiment and results**. All seems adequately addressed in the rebuttal.
- **No methodological foundation**. This point is not very well addressed, authors provide some evidence of smoother distributional gaps between adjacent steps, however this evidence still looks rather weak, since it is computed only for specific model, and has no theoretical grounds behind it. Area chair however believe that the paper is an empirical paper, based on **working** **heuristic** solution, which is good enough for publication.
- **Method stability and complexity**. This point is also weakly addressed during rebuttal, however area chair acknowledge that it is very hard to perform proper stability study with limited computational resources. Thus area chair believe, authors response is sufficient.
- **FFE strategy**. This strategy will indeed will not work for videos with scene changes, this is a limitation of the proposed method. This method however can work in some other scenarios, which area chair believe to be sufficient.
- **Unaddressed Risk of Error Reinforcement in Self-Distillation**. This point is related to stability, it is clear that with authors proposed setting there is not error reinforcement, or it does not affect results as much.

**Reviewer Scores:**

LHmL - stay the same.

MC9Q - stay the same.

wWot - stay the same (or increase to 5, given the authors efforts to address the concerns)

nsqN - stay the same

---

### Decision · Program_Chairs · 2026-01-26

Accept (Poster)